# From Indicators to Insights: Diversity-Optimized for Medical Series-Text Decoding via LLMs

**Xiyuan Jin**[1], **Jing Wang**[1,2,*] **Ziwei Lin**[1], **Qianru Jia**[1], **Yuqing Huang**[4],
**Xiaojun Ning**[1], **Zhonghua Shi**[5], **Youfang Lin**[1,3]

[1]School of Computer Science and Technology, Beijing Jiaotong University, Beijing, China
[2]Key Laboratory of Big Data & Artifcial Intelligence in Transportation
Ministry of Education, Beijing, China
[3]Beijing Key Laboratory of Traffic Data Mining and Embodied Intelligence, Beijing, China
[4]Beijing TongRen Hospital, Capital Medical University, Beijing, China
[5] Department of Neurosurgery, Department of Intensive Care Medicine,
Sanbo Brain Hospital, Capital Medical University, Beijing, China
`{xiyuanjin, wj, ziweilin, qianrujia, ningxj, yflin}@bjtu.edu.cn`
`{hyq9873, z.shi}@mail.ccmu.edu.cn`

## Abstract

Medical time-series analysis differs fundamentally from general ones by requiring specialized domain knowledge to interpret complex signals and clinical context. Large language models (LLMs) hold great promise for augmenting medical time-series analysis by complementing raw series with rich contextual knowledge drawn from biomedical literature and clinical guidelines. However, realizing this potential depends on precise and meaningful prompts that guide the LLM to key information. Yet, determining what constitutes effective prompt content remains non-trivial—especially in medical settings where signal interpretation often hinges on subtle, expert-defined decision-making indicators. To this end, we propose In-DiGO, a knowledge-aware evolutionary learning framework that integrates clinical signals and decision-making indicators through iterative optimization. Across four medical benchmarks, InDiGO consistently outperforms prior methods. The code is available at: `https://github.com/jinxyBJTU/InDiGO`.

## 1 Introduction

Medical time series analysis forms an essential foundation for continuous health monitoring, underpinning key applications such as sleep staging [11], cardiac arrhythmia detection [19], and mobility assessment [50]. Unlike generic time series analysis, medical data requires models to capture clinically meaningful signal patterns and incorporate domain-specific knowledge for trustworthy interpretation. Earlier studies integrating knowledge priors into model architectures and aligning loss functions with physiological semantics have yielded promising results [14, 33, 17].

In recent years, large language models (LLMs) have gained traction in time series analysis for their ability to integrate complementary knowledge through two key paradigms. The *implicit paradigm* builds on inductive biases embedded in pretrained architectures, harnessing LLMs' strength in compositional reasoning and contextual abstraction [36, 49]. Meanwhile, the *explicit paradigm* formulates expert knowledge as structured textual prompts, enabling direct and interpretable knowledge injection [12, 23, 24]. While both strategies offer notable advantages over traditional unimodal approaches, several key challenges remain in fully realizing the potential of LLMs for medical analysis.

---

*Corresponding author

39th Conference on Neural Information Processing Systems (NeurIPS 2025).

**C1: What kinds of promptable knowledge are most effective for decoding medical time series?**

The selection of textual content as a knowledge carrier varies across existing works. For example, Time-LLM [12] designs a template that includes dataset descriptions and sample statistics to guide prediction tasks, while AutoTimes [23] encodes timestamp information for token-wise prompting. KEDGN [24] uses variable-specific textual medical knowledge to model time-varying inter-variable dependencies. However, such promptable knowledge remains overly broad and lacks discriminative power even in common domain-informed tasks—let alone in expertise-intensive clinical domains.

In these contexts, a wealth of **decision-making indicators** actually remains overlooked. As shown in Figure 1, sleep waveforms like spindles and slow waves are key indicators for identifying sleep stages [3], while features such as RR intervals and QRS complex provide essential cues for cardiac assessment [8]. Although the extraction of these indicators via semi-automated tools may suffer from limited accuracy, they are easy to obtain and offer high utility for task, yet current LLM-assisted paradigms fail to leverage these crucial clues, missing important insights and losing a natural attribution pathway.

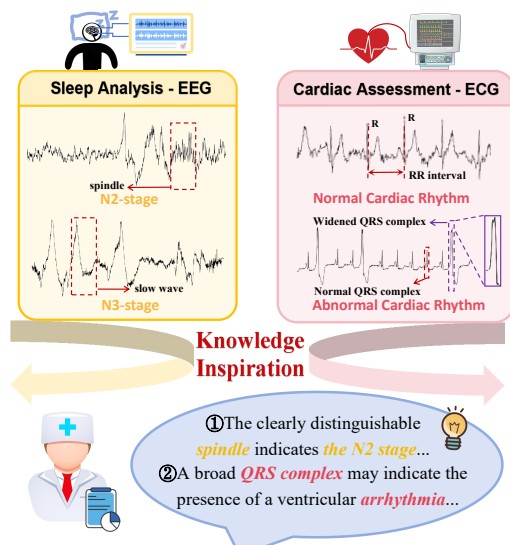

Figure 1: Task-relevant indicators provide critical cues for physiological state interpretation in sleep analysis and cardiac assessment.

**C2: How to robustly integrate time series and suboptimal text prompts?**

Despite growing efforts to integrate textual knowledge into time-series models, no unified strategy has emerged. A common approach directly concatenates textual and time-series representations as LLM input [4, 23], while others either enforce alignment constraints across modalities [36, 30], or use time-series features to query pretrained word embedding dictionaries for semantic enrichment [12, 22]. These strategies presuppose that textual inputs are sufficiently informative for the task. When this condition is unmet, there is no further mechanism to reconcile the misalignment between modalities, and the synergy between time-series and text collapses, offering little valuable information for the LLM to exploit—ultimately resulting in marginal performance gains and underutilization of its inferential capacity.

To this end, we propose **InDiGO—Indicator-informed Diversity-Guided Optimization**—a framework that integrates domain-informed decision indicators to enhance prompt robustness for medical time-series analysis. InDiGO incorporates alignment- and diversity-aware optimization to maximize synergy between text and time-series modalities, enabling more informative and resilient prompt learning while maintaining robustness to initial prompt variations across diverse clinical tasks.

- We present a theoretical analysis of joint time series–text decoding, revealing that existing limitations largely arise from biased estimation due to mismatched series-text pairs.
- We propose a mask-based importance sampling strategy grounded in indicator-informed prompts to approximate optimal series-text combinations through alignment and diversity optimization.
- Experiments on four real-world physiological datasets show that InDiGO outperforms state-of-the-art methods, with visualizations further demonstrating its strong interpretability.

## 2 Related Works

### 2.1 Knowledge-Empowered Medical Modeling

Prior to the rise of language models and multimodal learning, many studies incorporated prior knowledge through task-specific module design and task-agnostic loss design.

**Task-specific module design.** SleepHGNN [10] models the brain functional relationships across different sleep stages using heterogeneous graph convolution. L-SeqSleepNet [33] and SleepKD [21] both focus on modeling sleep transition rules between epochs, building upon the extraction of local features within each individual epoch.

**Task-agnostic loss design.** CLOCS [17], based on ECG acquisition scenarios, defines a family of patient-specific contrastive learning losses to guide the model in learning patient-centric invariances. Building on this, COMET [43] further strengthens the consistency and discriminative constraints at the trial and observation levels. ExpCLR [27] utilizes continuous expert features instead of view transformations to encourage consistency between the sample relationships and expert features. SmdCLR [13] extends InfoNCE [29] with multi-instance discrimination and semantic consistency regularization to alleviate false negatives and class imbalance in time-series contrastive learning.

## 2.2 LLM-Assisted Time Series Analysis

The emergence of pre-trained language models has revolutionized the knowledge-empowered modeling. Early works leverage LLMs' language priors for implicit knowledge modeling, while more recent efforts integrate textual content for explicit knowledge encoding.

**Implicit knowledge modeling.** PromptCAST [44] and LLMTIME [9] attempt to convert time series into text to exploit the zero-shot generalization capabilities inherent to LLMs. OneFitsAll [49] treats LLMs as universal sequence encoders by leveraging pre-trained self-attention to perform non-data-dependent operations, eliminating the need to convert time series into text. Following this, TEST [36] and $S^2$IP-LLM [30] attempted to align pre-trained dictionary tokens or text prototypes to better leverage the sequential modeling capabilities of LLMs.

**Explicit knowledge encoding.** TimeLLM [12] combines dataset information, task instructions, and input statistics into a template to provide a prompt prefix for time series, effectively activating specific tasks. TEMPO [4] introduces a semi-soft prompting strategy to provide component-specific prompts for different time series trend components. AutoTimes [23] transforms timestamps into textual prompts and integrates them with each series step as input, aligning with language models' autoregressive token modeling. KEDGN [24] converts the medical properties of variables into textual knowledge, enabling the language model to better capture the relationships between variables. TALON [37] further unifies time-series and textual modalities under a single large language model, enabling multimodal reasoning and generalizable forecasting across domains.

Despite these advances, existing approaches fall short in the context of physiological and healthcare data, leaving rich domain knowledge underexplored. Moreover, the paradigm of jointly decoding time series and text remains insufficiently explored and understood.

## 3 Preliminaries and Motivation

**Time-Series Text Joint Decoding.** In this study, each physiological sample can be represented by a triplet $(s_i, t_i, y_i)$, where $s_i \in \mathbb{R}^{C \times L}$ denotes the multichannel time series with $C$ channels and length $L$, $t_i$ is a variable-length descriptive text prompt, and $y_i \in \{1, 2, \ldots, K\}$ is the corresponding label among $K$ predefined categories. The joint decoding process can be formulated as:

$$P_{\text{LLM}}(Y = y_i | s_i, t_i) = \mathcal{N}(\mu_i(s_i, t_i; \theta), \sigma_i^2(s_i, t_i; \theta) * I) \tag{1}$$

where $\mu_i(s_i, t_i; \theta)$ and $\sigma_i^2(s_i, t_i; \theta)$ represent the predicted mean and variance respectively, and $I$ denotes the identity matrix, assuming independent output dimensions. By leveraging the conditional probability relationships, target $Y$ can be obtained through integration as:

$$P_{\text{LLM}}(Y) = \mathbb{E}_{(s,t) \sim P(s,t)} [P_{\text{LLM}}(Y|s,t)] \approx \frac{1}{n} \sum_{i=1}^{n} \mathbb{E}_{t \sim P(t|s_i)} [P_{\text{LLM}}(Y|s_i, t)] \tag{2}$$

where $P(t|s_i)$ is the distribution of possible text prompts conditioned on the series $s_i$, and the decoding process is achieved by integrating over the potential text distribution corresponding to all possible prompts for a given series $s_i$. Further, the optimal parameters $\theta^*$ for the LLM backbone as a joint decoding model can be estimated by Maximum Likelihood Estimation (MLE) as:

$$\theta^* = \text{argmin}_\theta - \sum_{i=1}^{n} \log P_{\text{LLM}}(Y = y_i) = \text{argmin}_\theta - \sum_{i=1}^{n} \log(\int P_{\text{LLM}}(y_i|s_i, t) P(t|s_i) dt) \tag{3}$$

This implies that the marginal likelihood (i.e., $\mathcal{M} = \int P_{\text{LLM}}(y_i|s_i,t)P(t|s_i)dt$) can be approximated by drawing a large number of samples from the potential distribution of all text prompts:

$$\hat{\mathcal{M}} = \mathbb{E}[P_{\text{LLM}}(y_i|s_i,t)P(t|s_i)] \approx \frac{1}{m}\sum_{j=1}^{m} P_{\text{LLM}}(y_i|s_i,t_j)P(t_j|s_i) \tag{4}$$

which is evidently impractical due to the immense computational and data requirements involved in such a process. In practical scenarios, using a finite number of text samples often leads to bias:

$$\text{Bias}(\hat{\mathcal{M}}) = \mathbb{E}[P_{LLM}(y_i|s_i,t)P(t|s_i)] - \int P_{LLM}(y_i|s_i,t)P(t|s_i)dt \tag{5}$$

where previous work usually leverages a specific text $t_i^j$ to essentially approximate the expectation of the posterior distribution $P(t|s_i)$. This can be interpreted in the form of a Laplace approximation:

$$\int P_{\text{LLM}}(y_i|s_i,t)P(t|s_i)dt \approx P_{\text{LLM}}(y_i|s_i,t_i^*)P(t_i^*|s_i)$$
$$t_i^* = \underset{t_i}{\text{argmax}}[\log P_{\text{LLM}}(y_i|s_i,t_i) + \log P(t_i|s_i)] \tag{6}$$

where $t_i^*$ represents the text that simultaneously maximizes both the likelihood function $P_{\text{LLM}}(y_i|s_i,t_i)$ and the posterior distribution $P(t_i|s_i)$, in other words, $t_i^*$ is the optimal text corresponding to the series $s_i$ under the target $y_i$. In the case of imprecise text $t$, such as rough statistical descriptions [12] or conceptual clinical knowledge [24], it is evident that this will cause severe bias.

In this paper, we propose an efficient solution based on importance sampling [26], aiming to quickly approximate the optimal text through multiple samplings from a simple distribution related to the initial prompt, thereby avoiding the significant bias risk caused by any single inaccurate text.

## 4 Methodology

The overall architecture of our model is illustrated in Figure 2. To better approximate the ideal marginal likelihood, we begin by crafting a contextually relevant text prompt by formalizing decision-making indicators extracted via semi-automated tools. Subsequently, we propose a masked Monte Carlo importance sampling (MCIS) mechanism, which supplants exhaustive prompt enumeration by iteratively sampling from a tractable proposal distribution. To foster deeper synergy between sampled prompts and time-series inputs, we introduce a heuristic optimization scheme that promotes informative and resilient prompt learning through relevance-based match alignment (MA) and diversity optimization (DO), while maintaining robustness to variations in initial prompt quality.

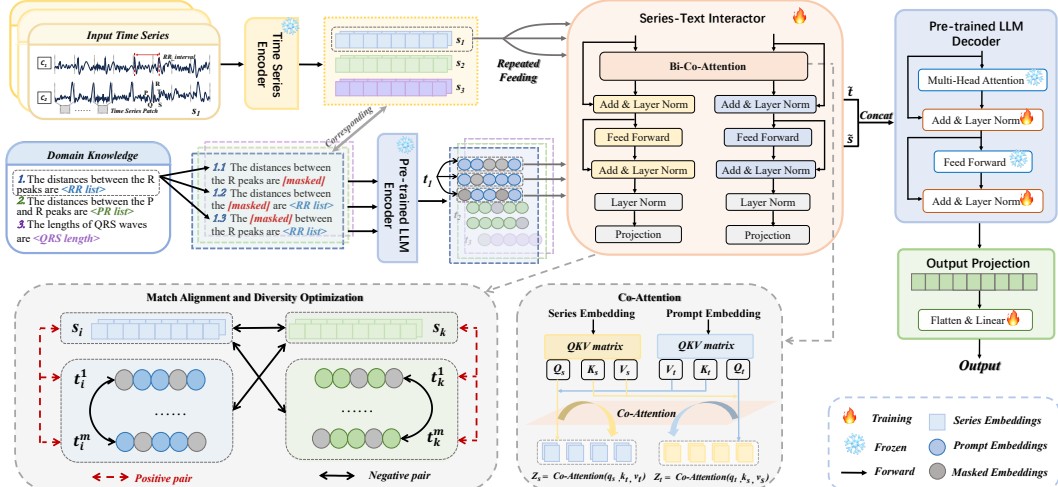

Figure 2: Overview of InDiGO. Given medical time series and initial texts, we apply mask-based importance sampling and pre-trained encoders to extract features. A series-text interactor captures relevance, followed by alignment and diversity optimization to identify the optimal combination.

## 4.1 Series Tokenization

Each input medical time series $s_i$ is passed through a pre-trained, frozen parameter time series encoder for feature extraction, where each series is initially divided into consecutive patches with length $L_p$. The total number of patches is given by $L_s = \lfloor \frac{L-L_p}{S} + 2 \rfloor$, where $S$ is the stride for the horizontal sliding operation. Once the patches are created, the pre-trained time series encoder will be leveraged to embed each patch in series $s_i = \{s_i^1, s_i^2, \ldots, s_i^{L_s}\}$ as:

$$\mathbf{s}_i = \textbf{SeriesEnc}(s_i^1, s_i^2, \ldots, s_i^{L_s}, [\text{CLS}^s]) \tag{7}$$

where $\mathbf{s}_i \in \mathbb{R}^{(L_s+1) \times d_s}$, and $[\text{CLS}^s]$ is the special category token added at the end of each series.

## 4.2 Indicator-Guided Prompt Prototype Construction

To provide as much complementary information as possible for time-series signals and facilitate faster approximation of the optimal textual description, we move beyond general dataset [12], timestamp-based descriptions [23] or conceptual clinical knowledge [24] and instead construct task-oriented prompts that incorporate key indicator-specific descriptions.

While it is challenging to exhaustively identify all task indicators, our goal is not to achieve completeness, but rather to extract relatively informative cues that complement input signals and enhance downstream task performance. As shown in Figure 3, an initial prompt prototype includes specific task instruction, a summary of signal statistics, and the presence or relationships of task-relevant indicators. For example, in sleep analysis, we detect slow and spindle waves in EEG signals, while in cardiac analysis, we extract rhythm-related features such as RR intervals, PR intervals, and QRS durations to support a better understanding of physiological states.

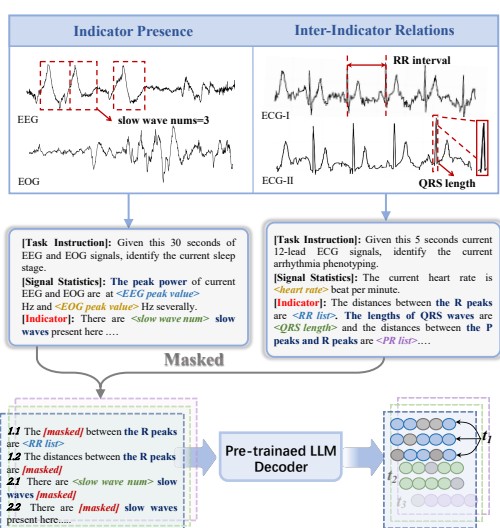

Figure 3: Scene-specific indicator prompting via semi-automated extraction.

These indicators are integrated into prompts through automated extraction, providing task-relevant insights and enabling strong potential for generalization to other expertise-intensive domains.

## 4.3 Masked Monte Carlo Importance Sampling

Based on the aforementioned indicator-guided prompts, we obtain an initial text sample $t_i^0$ corresponding to $s_i$, which serves as a coarse approximation of the optimal text $t_i^*$. However, even so, manually designed prompts inevitably introduce bias in the estimation of the marginal likelihood. To mitigate this limitation, we aim to construct and perform multiple importance samplings from a simple distribution $q(t|t_i^0)$ that is both computationally tractable and closer to the optimal distribution $t_i^*$, thereby replacing infeasible enumeration with sampling-driven surrogate approximation.

To be specific, for the estimation of the marginal likelihood $\mathcal{M}$, we can approximate it by performing multiple important sampling from a easy-tractable distribution $q(t|t_i^0)$:

$$\hat{\mathcal{M}} = \int P_{\text{LLM}}(y_i|s_i, t) q(t|t_i^0) \frac{P(t|s_i)}{q(t|t_i^0)} dt \approx \frac{1}{m} \sum_{j=1}^{m} P_{\text{LLM}}(y_i|s_i, t_i^j) \frac{P(t_i^j|s_i)}{q(t_i^j|t_i^0)} \tag{8}$$

where $q(t|t_i^0)$ is centered around a reference prompt $t_i^0$ and expected to cover the main support region of the target distribution $P(t|s_i)$. At this point, the optimal value of $t_i^*$ changes as follows:

$$t_i^* = \underset{t_i}{\text{argmax}}[\log P_{\text{LLM}}(y_i|s_i, t_i) + \log P(t_i|s_i) - \log q(t_i|t_i^0)] \tag{9}$$

In this way, the ideal marginal likelihood can be estimated efficiently rather than blindly attempting to replace it with a single textual prompt, which carries the risk of inaccuracy and mismatch.

**Masked Sampling.** We perform $m$ random masking operation iterations on the given text prompt $t_i^0$. Specifically, after tokenizing the textual prompts, we randomly select $r\%$ of the tokens at any position and replace them with the special **[Mask]** token, resulting in a masked sampling set $\mathcal{D}^{t_i}$:

$$\mathcal{D}^{t_i} = \{t_i^j | t_i^{j,1}, t_i^{j,2}, \dots, t_i^{j,L_{t_i}}; j = 1, \dots, m\} \tag{10}$$

where $L_{t_i}$ is the length of the prompt $t_i^0$ and each $t_i^j$ with a different random mask. Subsequently, we use a pre-trained text encoder to extract semantic representation of each $\mathbf{t}_i^j$ and repeat $m$ times:

$$\mathbf{t}_i^j = \textbf{TextEnc}([\text{CLS}^t], t_i^{j,1}, t_i^{j,2}, \dots, t_i^{j,l_i}, [\text{SEP}]) \tag{11}$$

where $\mathbf{t}_i^j \in \mathbb{R}^{(L_{t_i}+2) \times d_t}$, $[\text{CLS}^t]$ and $[\text{SEP}]$ are special tokens indicating the category and end of prompt $t_i^j$, respectively. This masked sampling and encoding process provides the initial (inaccurate) text prompt with an opportunity to infer and complete missing semantics from its surrounding context. To ensure consistency, both the text encoder and LLM decoder share the same architecture (e.g., GPT-2), using different layers from the front and back of the model.

## 4.4 Match Alignment and Diversity Optimization

By performing mask-based Monte Carlo sampling from a simple distribution, we obtain a set of candidate texts $\mathcal{D}^{t_i}$ that close to the optimal $t_i^*$, but their proximity still depends on the correlation between $q(t|t_i^0)$ and $P(t|s_i)$. Moreover, since the distribution $P(t|s_i)$ is intractable, directly minimizing the KL divergence between the sampling distribution and the true posterior is not feasible.

To address this, we introduce an indirect optimization strategy that provides a pathway for guiding the representations learning process of samples $\mathcal{D}^{t_i}$ to approximate the true posterior $P(t_i|s_i)$ through joint optimization with the downstream task objective $P_{\text{LLM}}(y_i|s_i, t_i)$, enabling the sampling process from $q(t|t_i^0)$ to evolve dynamically during training via modeling the series-text interaction understanding and a lightweight combination of alignment and diversity constraints.

**Series-Text Interaction.** We first model the correlation for each series-text pair $(s_i, t_i^j)$ using a co-attention mechanism in Series-Text Interactor (STI). Formally, we define the queries, keys and values matrices for both modalities. For each time series, the query, key and value are given by $\mathbf{q}_s = \mathbf{s}W_q^s$, $\mathbf{k}_s = \mathbf{s}W_k^s$ and $\mathbf{v}_s = \mathbf{s}W_v^s$, respectively. Similarly, for each text, the query, key and value are defined as $\mathbf{q}_t = \mathbf{t}W_q^t$, $\mathbf{k}_t = \mathbf{t}W_k^t$ and $\mathbf{v}_t = \mathbf{t}W_v^t$, where $W_q^s, W_k^s, W_v^s \in \mathbb{R}^{d_s \times d}$, and $W_q^t, W_k^t, W_v^t \in \mathbb{R}^{d_t \times d}$ are learnable weight matrices. The interaction between the time series $\mathbf{s}$ and the text $\mathbf{t}$ is modeled using a bidirectional co-attention mechanism, where each modality serves as both the query and the key/value for the other. Formally, we compute:

$$(\mathbf{z}_s, \mathbf{z}_t) = \text{BiCoAttn}(\mathbf{s}, \mathbf{t}) = \left( \text{softmax}\left( \frac{\mathbf{q}_s \mathbf{k}_t^\top}{\sqrt{d}} \right) \mathbf{v}_t, \ \text{softmax}\left( \frac{\mathbf{q}_t \mathbf{k}_s^\top}{\sqrt{d}} \right) \mathbf{v}_s \right) \tag{12}$$

where $\mathbf{z}_s \in \mathbb{R}^{(L_s+1) \times d}$ and $\mathbf{z}_t \in \mathbb{R}^{(L_t+2) \times d}$ denote the co-attended representations of the series and the text. These are later integrated into their original representations as updated $\mathbf{s}$ and $\mathbf{t}^j$.

**Alignment and Diversity Optimization for Evolving the Proposal Distribution.** Approximating the optimal $t_i^*$ involves both task likelihood and prompt posterior. Since the latter is intractable, we introduce a heuristic surrogate objective based on Bayes' theorem, replacing $P(t_i^j|s_i)$ with $P(s_i|t_i^j)$ and gradually guide the proposal distribution $q(t|t_i^0)$ toward the true posterior. We then design an alignment loss encourage precision and a diversity loss to promote coverage, jointly driving $q(t|t_i^0)$ to evolve toward the true posterior:

$$\mathcal{L}_{\text{objective}} = \mathbb{E}_{t_i^j \sim q(t|t_i^0)}[-\log P(s_i|t_i^j)] - H(q(t|t_i^0)) \tag{13}$$

where the first term encourages the sampled texts to be semantically informative with respect to $s_i$, while the second term promotes diversity by maximizing the entropy of the proposal distribution $q(t|t_i^0)$. This balance facilitates the discovery of more informative and diverse prompts that not only better approximate the unknown optimal prompt $t_i^*$, but also enhance resilience of inaccurate prompts to downstream tasks when integrated into fine-tuning.

Specifically, to further enhance semantic consistency between series $s_i$ and its sampled prompts $t_i^j$, we introduce a **match alignment loss** that explicitly aligns their representations:

$$\mathcal{L}_{\text{align}} = \mathbb{E}_{(s_i, t_i^j)}[-\log \frac{\exp(\text{sim}(\mathbf{s}_i, \mathbf{t}_i^j)/\tau)}{\sum_{k=1}^{N} \sum_{j=1}^{m} \exp(\text{sim}(\mathbf{s}_i, \mathbf{t}_k^j)/\tau)}] \tag{14}$$

Additionally, to encourage a broader exploration of the prompt space and reduce over-reliance on the initial prompt $t_i^0$, we incorporate a **diversity optimization objective** based on entropy:

$$\mathcal{L}_{\text{diverse}} = -H(q(t|t_i^0)) = \frac{1}{m} \sum_{j=1}^{m} \log q(t_i^j|t_i^0) \tag{15}$$

Together, these components enable the proposal distribution $q(t|t_i^0)$ to evolve during training in both a semantically aligned and exploration-rich manner, progressively approximating the true posterior.

**Joint Decoding and Output Projection.** For the multiple series-text pairs after STI modeling, we take their respective means $\tilde{\mathbf{s}}_i, \tilde{\mathbf{t}}_i$ and concatenate them, then feed the result into the LLM decoder and projection head for final inference, yielding the predicted target:

$$\hat{y}_i = \textbf{OutputProj}(\text{Flatten}(\textbf{Decoder}_{\text{LLM}}([\tilde{\mathbf{s}}_i, \tilde{\mathbf{t}}_i]))) \tag{16}$$

Finally the task loss (i.e., cross-entropy) collaborates with alignment and diversity objectives to jointly guide the model optimization and learning of proposal distribution as $\mathcal{L} = \mathcal{L}_{\text{align}} + \mathcal{L}_{\text{diverse}} + \mathcal{L}_{\text{task}}$.

## 5 Experiments

### 5.1 Datasets and Data Processing

**Datasets.** We evaluate InDiGO on four public medical healthcare datasets: **Sleep-EDF-20/78** (EEG/EOG, 100Hz) [16], **PTB-XL** (12-lead ECG, 500Hz) [41], and **UCI HAR** (wearable motion signals, 50Hz) [2]. Sleep-EDF-20/78 provides five sleep stages under AASM standards [3]; PTB-XL adopts two arrhythmia prototypes as established in prior work [45]; HAR supports six-class activity recognition from wearable sensors. More descriptions details please refer to Appendix A.

**Data Splitting.** We segment sleep recordings into 30s epochs and perform 5-fold subject-independent cross-validation (3:1:1 split). PTB-XL uses its standard 8:1:1 patient split, repeated over five seeds. HAR uses its official split, with the test set divided 1:1 into validation and test.

**Indicator Extraction.** For sleep staging, we annotate key indicators such as spindles and slow waves using A7 [18] and YASA [40]. For arrhythmia detection, we extract inter-wave features—RR, PR, QRS, and PP intervals—using NeuroKit2 [25]. HAR lacks distinct waveform structures, allowing us to evaluate model performance in the absence of explicit physiological indicators.

### 5.2 Experimental Settings and Baselines

The experiments are implemented by Pytorch framework. For Table 1 and Table 2, we report the mean and standard deviation values. Our model is trained by Adam optimizer with a learning rate lr=0.0003, a mask ratio of $r = 40\%$, and $m$=10 sampling times. We use BIOT [45], a pre-trained model tailored for physiological time series, as the time series encoder, and use the first/last 24 layers of GPT-2 XL [34] as the pre-trained text encoder and decoder, respectively. For details on pre-trained model configurations and hyperparameter choices, please refer to Table 9, 10 in Appendix B and C.

**Baselines.** We compare InDiGO with state-of-the-art methods, including general time-series representation models (TF-C [48], SimMTM [7], OneFitsAll [49], Time-LLM [12], KEDGN [24], MiniRocket [5], BIOT [45]), sleep stage classifiers (TinySleepNet [38], XSleepNet [32], L-SeqSleepNet [33], SleepHGNN [10], SleepKD [21], SleepDG [42], Brant-X [47]), and physiological signal decoders (SPaRCNet [15], ContraWR [46], CNN-Transformer [31], FFCL [20], ST-Transformer [35]).

### 5.3 Main Results

Tables 1 and 2 report the accuracy of InDiGO and baselines across three physiological tasks. InDiGO consistently outperforms all methods. General pre-trained models underperform due to their lack of

Table 1: 5-fold cross-validated average results for sleep stage classification.

| Methods | Sleep-EDF-20 | | | Sleep-EDF-78 | | |
|---|---|---|---|---|---|---|
| | Acc. | Macro F1 | Kappa | Acc. | Macro F1 | Kappa |
| TF-C [48] | 55.42 ±1.39 | 26.04 ±0.21 | 30.74 ±1.52 | 53.90 ±4.03 | 26.00 ±2.09 | 29.32 ±6.43 |
| SimMTM [7] | 66.91 ±1.89 | 53.21 ±1.95 | 53.25 ±2.02 | 63.06 ±2.67 | 57.07 ±2.13 | 53.07 ±3.42 |
| OneFitsAll [49] | 72.60 ±1.51 | 61.61 ±5.80 | 61.81 ±3.50 | 68.50 ±2.19 | 54.24 ±1.96 | 55.21 ±3.07 |
| Time-LLM [12] | 80.31 ±2.63 | 71.64 ±3.02 | 70.22 ±2.84 | 78.08 ±2.96 | 66.09 ±3.25 | 68.04 ±3.14 |
| KEDGN [24] | 74.89 ±3.86 | 64.29 ±3.36 | 64.90 ±5.46 | 70.34 ±1.85 | 58.59 ±2.74 | 57.47 ±2.56 |
| MiniRocket [5] | 81.60 ±1.55 | 72.82 ±2.01 | 72.79 ±1.96 | 78.36 ±1.93 | 70.18 ±2.35 | 69.46 ±2.46 |
| BIOT [45] | 81.86 ±4.41 | 75.29 ±4.47 | 75.14 ±6.00 | 77.15 ±3.04 | 69.36 ±4.13 | 68.26 ±4.36 |
| TinySleepNet [38] | 83.64 ±2.31 | 77.54 ±2.55 | 77.63 ±2.29 | 83.49 ±2.24 | 76.64 ±2.61 | 76.41 ±2.59 |
| XSleepNet [32] | 80.93 ±2.34 | 76.71 ±2.59 | 74.31 ±2.32 | 81.83 ±2.30 | 75.28 ±2.66 | 75.44 ±2.37 |
| L-SeqSleepNet [33] | 82.90 ±2.12 | 74.90 ±2.22 | 76.47 ±2.24 | 80.84 ±2.18 | 72.67 ±2.38 | 74.94 ±2.51 |
| SleepHGNN [10] | 81.15 ±1.96 | 72.88 ±2.17 | 73.35 ±2.16 | 77.35 ±2.13 | 69.56 ±2.39 | 68.65 ±2.41 |
| SleepKD [21] | 82.44 ±2.40 | 74.11 ±2.72 | 76.87 ±2.63 | 80.19 ±2.85 | 72.65 ±2.84 | 74.86 ±2.93 |
| SleepDG [42] | 81.92 ±2.27 | 74.74 ±2.53 | 76.43 ±2.47 | 79.95 ±2.42 | 72.21 ±2.59 | 74.16 ±2.68 |
| Brant-X [47] | 84.58 ±1.98 | 77.63 ±2.13 | 79.29 ±2.18 | 82.84 ±2.21 | 77.04 ±2.30 | 76.67 ±2.49 |
| **InDiGO** | **89.04** ±1.80 | **80.53** ±1.77 | **84.91** ±2.51 | **86.79** ±1.90 | **81.12** ±1.88 | **81.60** ±2.89 |

Table 2: Average performance on arrhythmia detection and human activity recognition tasks.

| Methods | PTB-XL | | | HAR | | |
|---|---|---|---|---|---|---|
| | BaAcc. | AUCPR | AUROC | BaAcc. | Kappa | Weighted F1 |
| TF-C [48] | 58.91 ±2.24 | 61.08 ±3.52 | 33.79 ±3.55 | 82.95 ±3.09 | 81.71 ±3.84 | 82.13 ±2.25 |
| SimMTM [7] | 64.13 ±2.16 | 67.91 ±2.80 | 42.95 ±4.98 | 88.53 ±0.52 | 86.42 ±0.63 | 88.40 ±0.51 |
| OneFitsAll [49] | 71.16 ±1.26 | 77.41 ±1.53 | 57.39 ±1.72 | 88.21 ±0.93 | 85.79 ±1.02 | 88.24 ±0.82 |
| Time-LLM [12] | 75.79 ±0.49 | 82.99 ±0.40 | 67.36 ±0.77 | 90.71 ±1.92 | 86.21 ±1.47 | 89.03 ±1.35 |
| KEDGN [24] | 74.64 ±1.42 | 80.06 ±1.83 | 66.73 ±2.49 | 89.50 ±0.28 | 86.14 ±0.33 | 88.74 ±0.24 |
| MiniRocket [5] | 81.79 ±0.34 | 87.79 ±0.13 | 75.03 ±0.19 | 91.68 ±0.53 | 89.58 ±0.25 | 91.32 ±0.69 |
| BIOT [45] | 84.21 ±0.30 | 92.21 ±0.75 | 76.59 ±0.76 | 94.61 ±1.34 | 93.51 ±1.60 | 94.58 ±1.36 |
| FFCL [20] | 70.34 ±0.52 | 70.88 ±0.53 | 51.27 ±0.51 | 85.19 ±1.48 | 82.16 ±1.77 | 85.08 ±1.38 |
| SPaRCNet [15] | 82.75 ±0.47 | 90.40 ±0.67 | 75.50 ±0.73 | 93.71 ±1.60 | 92.36 ±1.89 | 93.65 ±1.55 |
| ContraWR [46] | 75.32 ±5.61 | 75.49 ±1.64 | 52.58 ±11.90 | 90.68 ±1.64 | 88.79 ±2.01 | 90.55 ±1.82 |
| CNN-Transformer [31] | 66.50 ±4.59 | 71.75 ±5.58 | 49.96 ±9.36 | 86.90 ±8.39 | 82.73 ±9.53 | 83.52 ±11.66 |
| ST-Transformer [35] | 72.38 ±0.83 | 77.75 ±1.53 | 60.03 ±1.79 | 93.36 ±0.63 | 92.13 ±0.76 | 93.37 ±0.68 |
| **InDiGO** | **86.02** ±0.19 | **92.31** ±0.26 | **82.98** ±0.50 | **95.83** ±0.26 | **95.02** ±0.26 | **95.81** ±0.23 |

physiological signal awareness, while task-specific models benefit from prior knowledge integration. OneFitsAll, Time-LLM, and KEDGN despite leveraging knowledge encoding, offer limited gains, underscoring our method's strength. Notably, our advantage on the HAR task demonstrates that InDiGO remains effective even without task-specific waveform prompts.

## 5.4 Ablation Studies

Table 3: Ablation study of model design.

| Metrics | Sleep-EDF-20 | | PTB-XL | |
|---|---|---|---|---|
| | Macro F1 | Kappa | BaAcc. | AUCPR |
| w/o MCIS | 76.23 | 79.97 | 84.31 | 90.12 |
| w/o STI | 78.70 | 82.98 | 85.46 | 91.57 |
| w/o MA | 78.31 | 82.18 | 85.89 | 91.25 |
| w/o DO | 77.12 | 80.33 | 84.52 | 90.54 |
| **Full** | 80.53 | 84.91 | 86.02 | 92.31 |

To better understand the contribution of each component, we conduct ablation studies as summarized in Table 3. We can observe a consistent performance decline across all metrics when any of the key modules is ablated, highlighting their individual importance. w/o Masked Monte Carlo Importance Sampling (MCIS) shows the most significant drop, highlighting the critical role of masked importance sampling in enhancing robustness against inaccurate prompts. w/o Series-Text Interaction (STI) module also causes a notable degradation, which underscores the necessity of cross-modal interaction in capturing complementary information from series and text. w/o Match Alignment (MA) and w/o Diversity Optimization (DO) lead to moderate declines, highlighting their contribution to enhancing textual diversity, which in turn enables more effective activation of LLM's potential through joint fine-tuning.

## 5.5 Low-Resource Generalization: Few-Shot and Zero-Shot Settings

We evaluate InDiGO in low-resource settings, including few-shot and zero-shot scenarios. As shown in Tables 4 and 5, it consistently achieves the best results across all datasets, demonstrating

strong generalization and robustness. This improvement can be attributed to the indicator-guided prompting and mask-based diversity sampling, which jointly expand the textual representation space and strengthen series–prompt alignment, enabling effective adaptation under limited supervision.

Table 4: Performance Comparison on Few-Shot Learning Tasks (5% Training Data).

| Methods | SimMTM | | OneFitsAll | | Time-LLM | | KEDGN | | BIOT | | InDiGO | |
|---|---|---|---|---|---|---|---|---|---|---|---|---|
| | Acc. | Macro F1 | Acc. | Macro F1 | Acc. | Macro F1 | Acc. | Macro F1 | Acc. | Macro F1 | Acc. | Macro F1 |
| EDF-20 | 63.91 | 47.08 | 60.81 | 53.81 | 59.21 | 50.37 | 66.65 | 52.79 | 65.72 | 51.49 | **75.40** | **69.63** |
| EDF-78 | 62.06 | 42.72 | 60.28 | 52.08 | 66.47 | 55.14 | 66.20 | 51.53 | 66.38 | 51.37 | **74.81** | **68.63** |
| PTB-XL | 55.44 | 48.91 | 61.01 | 55.55 | 64.79 | 57.38 | 62.99 | 56.25 | 65.38 | 60.99 | **72.46** | **67.06** |
| HAR | 55.75 | 50.30 | 52.55 | 47.20 | 55.40 | 48.40 | 58.92 | 57.69 | 67.18 | 63.46 | **76.79** | **74.60** |

Table 5: Performance Comparison on Few-Shot Learning Tasks (10% Training Data).

| Methods | SimMTM | | OneFitsAll | | Time-LLM | | KEDGN | | BIOT | | InDiGO | |
|---|---|---|---|---|---|---|---|---|---|---|---|---|
| | Acc. | Macro F1 | Acc. | Macro F1 | Acc. | Macro F1 | Acc. | Macro F1 | Acc. | Macro F1 | Acc. | Macro F1 |
| EDF-20 | 64.03 | 48.78 | 64.10 | 56.96 | 67.42 | 53.55 | 69.37 | 57.16 | 68.99 | 56.34 | **76.70** | **70.83** |
| EDF-78 | 62.82 | 44.49 | 63.98 | 53.21 | 68.11 | 57.65 | 66.97 | 53.31 | 67.00 | 53.39 | **75.30** | **69.09** |
| PTB-XL | 56.15 | 53.24 | 67.14 | 56.39 | 65.01 | 58.82 | 64.92 | 57.14 | 70.78 | 64.57 | **75.86** | **70.30** |
| HAR | 57.96 | 53.38 | 62.38 | 61.32 | 63.76 | 62.78 | 60.69 | 59.20 | 69.42 | 68.98 | **79.45** | **79.45** |

We further conduct zero-shot transfer experiments between Sleep-EDF-20 and Sleep-EDF-78 by excluding the first 20 overlapping subjects from Sleep-EDF-78, as shown in Table 6. Even without target-domain labels, InDiGO surpasses baselines, showing that indicator-guided prompting and Monte Carlo sampling boost representation diversity and low-resource adaptability.

Table 6: Performance Comparison on Zero-Shot Learning Tasks.

| Methods | SimMTM | | OneFitsAll | | Time-LLM | | KEDGN | | BIOT | | InDiGO | |
|---|---|---|---|---|---|---|---|---|---|---|---|---|
| | Acc. | Macro F1 | Acc. | Macro F1 | Acc. | Macro F1 | Acc. | Macro F1 | Acc. | Macro F1 | Acc. | Macro F1 |
| EDF-20 → EDF-78 | 54.49 | 44.18 | 64.57 | 56.57 | 63.97 | 55.22 | 63.87 | 51.82 | 64.55 | 53.11 | **73.57** | **68.09** |
| EDF-78 → EDF-20 | 64.37 | 54.93 | 76.09 | 68.70 | 75.49 | 67.55 | 71.41 | 60.19 | 73.30 | 63.24 | **76.86** | **72.94** |

## 5.6 Efficiency Analysis

We conducted an efficiency comparison on the PTB-XL dataset to evaluate the computational cost of InDiGO against several representative baselines. As shown in Table 7, SimMTM exhibits the highest training cost, primarily due to its masked sampling mechanism and multi-scale contrastive objective. TimeLLM incurs substantial inference latency because it generates a separate textual prompt for each channel, introducing repeated computation. KEDGN suffers from the limited parallelism of its recurrent backbone, leading to

Table 7: Efficiency Evaluation: Training and Inference Overhead.

| Model Name | Training(s) | Inference(s) |
|---|---|---|
| SimMTM (m=3) | 2333.13 | 38.73 |
| TimeLLM | 1742.10 | 237.05 |
| KEDGN | 2321.37 | 241.27 |
| **InDiGO (m=5)** | 632.98 | 127.91 |
| **InDiGO (m=10)** | 788.79 | 144.23 |
| **InDiGO (m=20)** | 1118.17 | 175.12 |

longer execution times. In contrast, our method maintains high efficiency across varying prompt sample times ($m$ = 5, 10, 20). While increasing $m$ leads to a slight rise in training and inference time, InDiGO consistently outperforms the baselines in computational efficiency.

## 5.7 Diversity Enhancement via DO: Prompt Similarity Distribution Analysis

As shown in Figure 4, DO significantly shifts the distribution toward lower similarity values across all datasets, indicating a broader semantic spread among the sampled prompts. This validates that DO promotes greater lexical and semantic diversity, thereby enriching the representational space of textual cues. Such diversity enables the model to explore a wider range of series-text alignments during fine-tuning, improving generalization especially under weak or partially informative prompts.

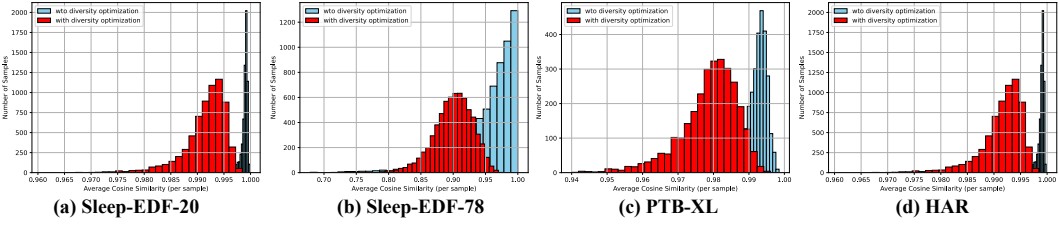

(a) Sleep-EDF-20    (b) Sleep-EDF-78    (c) PTB-XL    (d) HAR

Figure 4: Distribution of prompt similarities with/without Diversity Optimization across four datasets.

## 5.8 Prompt Robustness under Alignment and Diversity Objectives

Here, we further provide experimental validation of the effectiveness and robustness of indicator-guided prompts, match alignment (MA), and diversity optimization (DO).

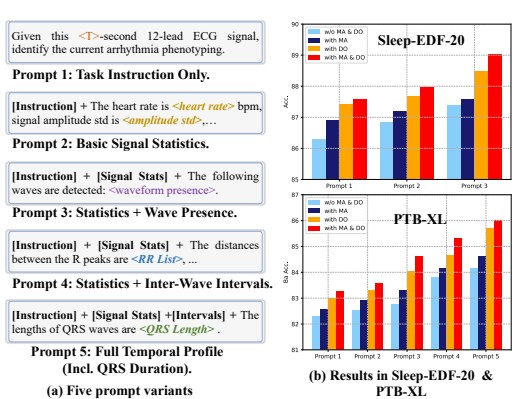

(a) Five prompt variants

(b) Results in Sleep-EDF-20 & PTB-XL

Figure 5: Different prompt designs.

Specifically, we progressively refine the prompts, starting from general instruction of the task and signal statistics, and incrementally incorporating task-relevant domain indicators to enhance their specificity. We then evaluate the model under all combinations of match alignment and diversity optimization, including the baseline without either strategy. As shown in Figure 5 (results for Sleep-EDF-78 are in Appendix F), our MA-DO optimization framework ensures robust adaptation across varying prompt qualities. Notably, even under limited or generic prompt conditions, MA and DO offer stable gains. More importantly, once key indicator cues are partially introduced, these strategies effectively expand the semantic utility of prompts—facilitating alignment between textual hints and task objectives, and substantially improving model generalization.

## 5.9 Model Interpretation Analysis of Series-Text Interactions

We present a case study on Sleep-EDF-20, as illustrated in Figure 6. We visualize the co-attention scores between series and text prompts throughout the network's optimization process. The top 3 subplots show the evolution of the network's learning of the series-text correlation: from initial misalignment in **(a)** to indiscriminate attention in the middle stages in **(b)**, and finally to a more refined, discriminative focus on text prompts as the network matures in **(c)**. The patches with higher attention scores correspond to specific spindles in the sleep EEG, indicating that our series-text interaction process is progressively guided by localized waveform indicators. Moreover, the indicator-related sentences in the textual prompts gradually play a more significant role, while the statistical feature parts become less influential during network optimization. This

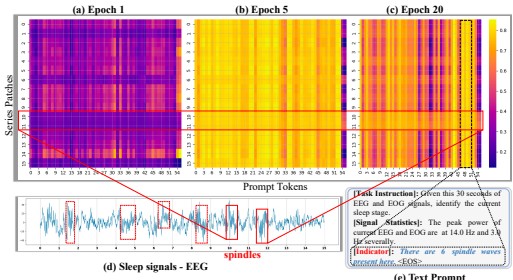

Figure 6: A showcase of series-text correlation with network training. (a)-(c) represent the co-attention scores of the series-text pair, (d) and (e) represent the series and text samples.

further demonstrates that, through informative indicator-related prompts and the dynamic evolution guided by MA-DO sampling, our model enhances the mutual understanding between time series and text, allowing the embedded knowledge to be effectively interpreted by the model.

## 6 Conclusion

In this paper, we propose the Indicator-informed Diversity-Guided Optimization framework (In-DiGO), a knowledge-aware method tailored for multimodal modeling of medical time-series analysis. While our current approach requires relatively structured indicator-to-text conversion—often relying on domain expertise—we demonstrate that InDiGO significantly lowers the barrier for clinical professionals to participate in LLM-driven multimodal analysis, offering a promising step toward broader adoption of foundation models in medical time-series applications.

## Acknowledgments

This work was supported by the National Natural Science Foundation of China (No. 62576027) and the Fundamental Research Funds for the Central Universities under Grant No. 2025JBMC030.

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

# A    Details of Datasets and Experimental Settings

## A.1    More for Datasets and Processings

We evaluate the proposed InDiGO on four diverse physiological datasets to assess its generalizability across modalities. The dataset statistics are summarized in Table 8.

Table 8: Dataset statistics.

| Datasets | Sleep-EDF-20 | Sleep-EDF-78 | PTB-XL | HAR |
|---|---|---|---|---|
| # Samples | 42,308 | 195,479 | 65,511 | 10,299 |
| # Rate | 100 | 100 | 500 | 50 |
| # Channels | 1 EEG,1 EOG | 1 EEG,1 EOG | 12 ECG leads | 9 coordinates |
| # Duration | 30 seconds | 30 seconds | 5 seconds | 2.56 seconds |
| # Max Observation Length | 3000 | 3000 | 2400 | 128 |

- **Sleep-EDF-20** & **Sleep-EDF-78** [2]: Contain overnight polysomnographic recordings with EEG Fpz-Cz and horizontal EOG signals sampled at 100 Hz. Sleep stages are annotated based on AASM rules into five categories: Wake, N1, N2, N3, and REM. Sleep-EDF-20 includes 39 recordings from 20 subjects; Sleep-EDF-78 contains 153 recordings from 78 participants aged 25–101. We segment recordings into 30-second epochs, and retain corresponding sleep labels. We adopt a 5-fold subject-independent cross-validation setup. Each fold divides subjects into training, validation, and test sets with a 3:1:1 ratio.

- **PTB-XL** [3]: A large-scale 12-lead ECG dataset with 21,837 10-second recordings from 18,885 patients. For all signals, we use the original standard set of 12 leads (I, II, III, AVL, AVR, AVF, V1, ..., V6) with reference electrodes on the right arm. Each ECG record is labeled using standardized SCP-ECG statements across diagnostic, form, and rhythm classes. Recordings are sampled at 500 Hz. We use the standard 80%/10%/10% patient-level split and repeat experiments five times with different random seeds. Each ECG is associated with multiple diagnostic labels (up to 27 classes). Following BIOT [45], we group arrhythmia-related diagnoses into a binary label: a record is considered positive if at least one arrhythmia-type diagnosis is present.

- **HAR** [4]: Contains tri-axial accelerometer and gyroscope signals (50 Hz) collected from 30 individuals performing six daily activities while wearing a waist-mounted Samsung Galaxy S2 smartphone. Activities include walking, walking upstairs/downstairs, sitting, standing, and lying down. We directly use the pre-segmented time windows and activity labels. The dataset is used as-is for 6-class human activity recognition to validate our method's adaptability to non-waveform physiological time series.

We use 1 RTX A6000 48GB GPU to train models for 200 epochs with a batch size of 256. Learning rate and weight decay are grid-searched to the optimal.

## A.2    Evalution metrics

Below, we provide the definitions of some of the metrics used in our study.

- **Macro F1** is the arithmetic mean of the F1 scores computed for each class, treating each class equally regardless of its frequency in the dataset.

- **Coken's Kappa** is a statistical measure used to assess the level of agreement between two raters (or systems) when classifying items into mutually exclusive categories. It considers both the observed agreement (the proportion of times the raters agree) and the expected agreement (the agreement that would occur by chance).

---

[2] https://www.physionet.org/content/sleep-edfx/1.0.0/
[3] https://physionet.org/content/ptb-xl/1.0.1/
[4] https://archive.ics.uci.edu/dataset/240/human+activity+recognition+using+smartphones

- **Balanced Accuracy** is a metric used to evaluate the performance of a classification model, especially when the dataset is imbalanced. It calculates the average of the sensitivity (true positive rate) and specificity (true negative rate) to ensure that both classes are treated equally.
- **AUC-PR** is the area under the precision recall (PR) curve for binary classification task.
- **AUROC** is the area under the ROC curve, summarizing the ROC curve into an single number that describes the performance of a model for multiple thresholds at the same time.

# B  Selection of Pretrained Models

## B.1  Pretrained Series Encoder

Due to the variability of physiological time-series scenarios, where different diseases or treatment processes involve distinct data collection paradigms, there is currently no mature, widely recognized multi-modal physiological time-series foundation model. In the context of this study, we utilize BIOT [45], a model specifically pre-trained on EEG and ECG data.

**Sleep stage classification.** We initialize our time series encoder using the pre-trained model provided by the official release, which was trained on six EEG datasets.

**Arrhythmias phenotype detection.** Since the official code of BIOT [45] does not provide pre-trained models for ECG tasks, we follow the strategy outlined in its paper and perform unsupervised pre-training on five cardiology subsets [1] using 12-lead ECG data.

## B.2  Pretrained LLM

Table 9: The classification performance of different pretrained LLMs on Sleep-EDF-20 and PTB-XL.

| Model Name | Parameters | Layers | $d_{model}$ | Sleep-EDF-20 | | | | | PTB-XL | | |
|---|---|---|---|---|---|---|---|---|---|---|---|
| | | | | Acc. | Sens | Spec | Macro F1 | Kappa | BaAcc. | AUCPR | AUROC |
| BERT | 110M | 12 | 768 | 85.73 | 76.98 | 95.72 | 75.39 | 80.98 | 84.15 | 89.97 | 80.73 |
| GPT-2 Small | 124M | 12 | 768 | 86.69 | 77.41 | 96.90 | 76.04 | 81.68 | 84.97 | 90.84 | 81.48 |
| GPT-2 Medium | 355M | 24 | 1024 | 88.10 | 78.42 | 97.24 | 77.51 | 83.59 | 84.97 | 90.95 | 81.79 |
| GPT-2 Large | 774M | 36 | 1280 | 88.55 | 80.12 | 97.35 | 79.29 | 84.22 | 85.44 | 91.09 | 81.33 |
| GPT-2 XL | 1.5B | 48 | 1600 | 89.04 | 81.32 | 97.44 | 80.53 | 84.91 | 86.02 | 92.31 | 82.98 |
| LLaMA(8) | 1.71B | 8 | 4096 | 86.55 | 78.74 | 95.93 | 77.22 | 81.24 | 83.95 | 90.13 | 80.84 |
| LLaMA(32) | 7B | 32 | 4096 | 88.13 | 79.29 | 96.86 | 79.15 | 82.45 | 84.94 | 90.82 | 81.35 |

We evaluated the classification performance of several pre-trained LLMs as the language encoder-decoder in **InDiGO**, including four versions of GPT-2 [34], BERT [6], and two variants of LLaMA [39], as summarized in Table 9. Overall, different LLMs yield comparable performance across both tasks. We attribute this to the nature of our physiological time series classification task, where the textual prompts are relatively concise and structurally simple, placing limited demand on the encoder's capacity for deep semantic reasoning.

Notably, increasing model size (e.g., among GPT-2 variants) does not bring substantial gains, while the performance of LLaMA models shows no consistent advantage despite their high dimensionality—possibly due to optimization difficulties during training. These observations further confirm that the primary challenge in our setting lies not in modeling complex text, but in enhancing the mutual understanding between time series and prompts through effective series-text alignment.

# C  Selection of optimal hyper-parameters

We provide diagnostic experiments on the selection of hyperparameters in the InDiGO, as shown in Table 10. It is important to note that when adjusting a specific parameter, the remaining parameters are set to the same values as those of the optimal result.

**Layers of LLM Encoder and Decoder.** Considering the joint series-text decoding process in this paper and the encoding process of initial textual prompts, the text encoder and decoder used in

Table 10: Diagnostic experiments of our model on Sleep-EDF-20 and PTB-XL.

| Variants | | Sleep-EDF-20 | | | | | PTB-XL | | |
|---|---|---|---|---|---|---|---|---|---|
| | | Acc. | Sens | Spec | Macro F1 | Kappa | BaAcc. | AUCPR | AUROC |
| **Full Model** | $L_{Enc}=24, L_{Dec}=24$ $r=40\%, m=10$ | **89.04** | **81.32** | **97.44** | **80.53** | **84.91** | **86.02** | **92.31** | **82.98** |
| **Layer of LLM Enc-Dec** | $L_{Enc}=6, L_{Dec}=42$ | 87.62 | 79.13 | 96.91 | 78.03 | 84.12 | 84.72 | 90.01 | 80.73 |
| | $L_{Enc}=12, L_{Dec}=36$ | 88.00 | 79.86 | 97.21 | 78.84 | 83.51 | 85.27 | 90.62 | 81.04 |
| | $L_{Enc}=18, L_{Dec}=30$ | 88.62 | 79.75 | 97.38 | 78.38 | 84.28 | 85.57 | 91.37 | 82.92 |
| | $L_{Enc}=30, L_{Dec}=18$ | 88.93 | 79.59 | 97.45 | 78.62 | 84.71 | 85.10 | 90.97 | 81.63 |
| | $L_{Enc}=36, L_{Dec}=12$ | 88.65 | 79.34 | 97.41 | 78.17 | 84.32 | 84.83 | 90.12 | 80.79 |
| | $L_{Enc}=42, L_{Dec}=6$ | 87.89 | 79.43 | 97.20 | 77.96 | 83.35 | 84.31 | 90.79 | 81.27 |
| **Mask Ratio** | $r=20\%$ | 86.49 | 74.77 | 96.88 | 73.17 | 81.32 | 85.26 | 90.60 | 80.35 |
| | $r=60\%$ | 88.73 | 79.80 | 97.40 | 79.11 | 84.45 | 85.34 | 90.65 | 81.53 |
| | $r=80\%$ | 87.93 | 80.32 | 97.18 | 79.14 | 83.44 | 84.06 | 90.41 | 81.28 |
| **Sampling times** | $m=5$ | 88.20 | 79.53 | 97.28 | 78.49 | 83.73 | 84.41 | 90.21 | 80.24 |
| | $m=15$ | 89.34 | 79.78 | 97.55 | 78.93 | 85.28 | 85.36 | 90.69 | 80.82 |
| | $m=20$ | 89.18 | 79.45 | 97.54 | 78.26 | 85.03 | 85.39 | 90.44 | 81.03 |
| | $m=30$ | 89.46 | 80.57 | 97.55 | 80.52 | 85.46 | 85.38 | 90.79 | 81.23 |

InDiGO are derived from different layers of GPT-2. We divide the 48 hidden layers of GPT-2 XL into several groups based on their sequential order. Since we freeze the text encoder and fine-tune the text decoder, different layer combinations can impact the optimization speed of the network. However, a more critical issue lies in the alignment condition. Mismatched layer configurations between the text encoder and decoder may lead to misalignment, causing the representations of the series and text under the alignment loss constraint to lose their intended meaning. From the experimental results, we observe that when the language model layers are balanced between the encoder and decoder, our series-text joint decoding achieves the best performance on downstream tasks. This conclusion also holds true across different scales of GPT-2.

**Mask Ratio.** In the masked Monte Carlo importance sampling, we achieve sampling from a simple distribution by applying token-wise random masking to the initial waveform-related text prompts. During the adjustment of the masking probability, we observed that with a fixed number of samples, a lower masking rate limited the diversity of the samples, while a higher masking rate caused distortion in the generated text. Therefore, the optimal masking ratio, as experimentally proven, is 40%, though it also depends on the number of subsequent samples.

**Sampling Times.** In fact, the number of samples is closely related to the masking ratio. When the masking ratio is low, fewer sampling iterations do not effectively approximate the optimal text. When the masking ratio is moderate, adjusting the number of samples still doesn't necessarily lead to better results with too many iterations. Therefore, considering the training cost, we select 10 sampling iterations as the optimal balance.

## D  Full Results

Due to space limitations and clarity considerations in the main text, we did not report the full set of evaluation metrics for the sleep staging task. For completeness, we present the detailed results in Table 11. Specifically, we additionally report sensitivity and specificity, which provide further insights into stage-wise classification performance.

## E  Statistical Significance Analysis of Model Performance

Table 12 presents the statistical significance results of model performance comparisons against InDiGO across four benchmark datasets using MANOVA tests [28]. From the table, we observe that InDiGO consistently and significantly outperforms all baseline models across all datasets. Most comparisons reach the highest level of significance, suggesting that InDiGO's superiority is not only consistent but also statistically robust. Particularly, on the PTB-XL dataset, all models show highly significant differences, indicating that InDiGO's improvements are most prominent in complex physiological signal domains. Similarly, on HAR, all models except BIOT achieve *** significance, while BIOT still maintains a ** level, reflecting InDiGO's strong performance even against recent competitive baselines.

Table 11: Average performance on the sleep stage classification task.

| Methods | Sleep-EDF-20 | | | | | Sleep-EDF-78 | | | | |
|---|---|---|---|---|---|---|---|---|---|---|
| | Acc. | Sens. | Spec. | Macro F1 | Kappa | Acc. | Sens. | Spec. | Macro F1 | Kappa |
| TF-C [48] | 55.42 ±1.39 | 31.52 ±1.09 | 86.07 ±0.39 | 26.04 ±0.21 | 30.74 ±1.52 | 53.90 ±4.03 | 31.35 ±2.40 | 85.80 ±1.34 | 26.00 ±2.09 | 29.32 ±6.43 |
| SimMTM [7] | 66.91 ±1.89 | 53.47 ±1.58 | 90.61 ±1.63 | 53.21 ±1.95 | 53.25 ±2.02 | 63.06 ±2.67 | 59.12 ±3.88 | 91.21 ±1.56 | 57.07 ±2.13 | 53.07 ±3.42 |
| OneFitsAll [49] | 72.60 ±1.51 | 63.50 ±8.36 | 92.76 ±1.12 | 61.61 ±5.80 | 61.81 ±3.50 | 68.50 ±2.19 | 56.58 ±4.16 | 91.34 ±0.86 | 54.24 ±1.96 | 55.21 ±3.07 |
| Time-LLM [12] | 80.31 ±2.63 | 76.53 ±3.15 | 94.53 ±2.95 | 71.64 ±3.02 | 70.22 ±2.84 | 78.08 ±2.96 | 67.44 ±3.73 | 94.13 ±3.01 | 66.09 ±3.25 | 68.04 ±3.14 |
| KEDGN [24] | 74.89 ±3.86 | 64.60 ±3.68 | 93.39 ±1.26 | 64.29 ±3.36 | 64.90 ±5.46 | 70.34 ±1.85 | 57.70 ±2.01 | 92.31 ±0.49 | 58.59 ±2.74 | 57.47 ±2.56 |
| MiniRocket [5] | 81.60 ±1.55 | 72.63 ±1.80 | 95.15 ±1.12 | 72.82 ±2.01 | 72.79 ±1.96 | 78.36 ±1.93 | 69.76 ±2.44 | 94.08 ±1.76 | 70.18 ±2.35 | 69.46 ±2.46 |
| BIOT [45] | 81.86 ±4.41 | 75.98 ±3.66 | 95.12 ±1.31 | 75.29 ±4.47 | 75.14 ±6.00 | 77.15 ±3.04 | 70.85 ±2.87 | 93.94 ±0.76 | 69.36 ±4.13 | 68.26 ±4.36 |
| TinySleepNet [38] | 83.64 ±2.31 | **81.60** ±2.60 | 96.05 ±2.08 | 77.54 ±2.55 | 77.63 ±2.29 | 83.49 ±2.24 | 80.25 ±2.65 | 96.02 ±2.11 | 76.64 ±2.61 | 76.41 ±2.59 |
| XSleepNet [32] | 80.93 ±2.34 | 75.78 ±2.21 | 94.79 ±2.54 | 76.71 ±2.59 | 74.31 ±2.32 | 81.83 ±2.30 | 80.50 ±2.28 | 95.74 ±2.58 | 75.28 ±2.66 | 75.44 ±2.37 |
| L-SeqSleepNet [33] | 82.90 ±2.12 | 78.42 ±2.25 | 95.86 ±2.00 | 74.90 ±2.22 | 76.47 ±2.24 | 80.84 ±2.18 | 72.75 ±2.54 | 95.19 ±2.34 | 72.67 ±2.38 | 74.94 ±2.51 |
| SleepHGNN [10] | 81.15 ±1.96 | 74.23 ±2.10 | 94.93 ±1.96 | 72.88 ±2.17 | 73.35 ±2.16 | 77.35 ±2.13 | 69.94 ±2.48 | 94.04 ±2.02 | 69.56 ±2.39 | 68.65 ±2.41 |
| SleepKD [21] | 82.44 ±2.40 | 78.20 ±2.54 | 94.78 ±2.34 | 74.11 ±2.72 | 76.87 ±2.63 | 80.19 ±2.85 | 72.95 ±2.88 | 94.95 ±2.69 | 72.65 ±2.84 | 74.86 ±2.93 |
| SleepDG [42] | 81.92 ±2.27 | 79.12 ±2.35 | 95.75 ±2.68 | 74.74 ±2.53 | 76.43 ±2.47 | 79.95 ±2.42 | 73.31 ±2.41 | 93.57 ±2.63 | 72.21 ±2.59 | 74.16 ±2.68 |
| Brant-X [47] | 84.58 ±1.98 | 80.18 ±2.23 | 96.36 ±1.89 | 77.63 ±2.13 | 79.29 ±2.18 | 82.84 ±2.21 | 81.85 ±2.42 | 95.91 ±2.08 | 77.04 ±2.30 | 76.67 ±2.49 |
| **InDiGO** | **89.04** ±1.80 | 81.32 ±1.64 | **97.44** ±0.44 | **80.53** ±1.77 | **84.91** ±2.51 | **86.79** ±1.90 | **81.86** ±1.63 | **96.71** ±0.59 | **81.12** ±1.88 | **81.60** ±2.89 |

On Sleep-EDF-20 and Sleep-EDF-78, although all models are still significantly outperformed by InDiGO, the significance levels are slightly more diverse. This reflects the relatively closer performance gap among top models in sleep staging tasks, especially between InDiGO and MiniRocket or BIOT. Nonetheless, even in these settings, the improvements by InDiGO are statistically justified.

Overall, this analysis reaffirms that InDiGO offers a substantial and statistically significant performance gain across varied time-series datasets, underscoring its effectiveness and generalizability.

Table 12: Statistical significance of model performance compared to InDiGO on each dataset. Stars indicate significance levels from MANOVA tests [28]. * $p < 0.05$, ** $p < 0.01$, *** $p < 0.001$

| Model / Dataset | TF-C | SimMTM | OneFitsAll | Time-LLM | KEDGN | MiniRocket | BIOT |
|---|---|---|---|---|---|---|---|
| Sleep-EDF-20 | *** | *** | *** | ** | *** | ** | * |
| Sleep-EDF-78 | *** | *** | *** | ** | *** | ** | ** |
| PTB-XL | *** | *** | *** | *** | *** | *** | *** |
| HAR | *** | *** | *** | *** | *** | *** | ** |

# F  Robustness Across Prompt Variants on Sleep-EDF-78

To further validate the generalizability of our approach, we replicate the prompt sensitivity study on the Sleep-EDF-78 dataset. We consider three types of indicator-guided prompts with increasing specificity: (i) prompts describing only the sleep staging task, (ii) prompts augmented with statistical characteristics of sleep signals, (iii) prompts additionally incorporating references to sleep waveform patterns.

As shown in Figure 7, results on Sleep-EDF-78 mirror the trends observed in the main datasets. Specifically, our MA-DO framework consistently improves model performance under less informative prompts by broadening the effective representational space of textual inputs. Notably, once waveform-based indicators are introduced, performance gains become more substantial—highlighting the model's improved ability to leverage semantically rich cues through alignment and diversity enhancement.

# G  Additional Interpretation Analysis: Slow-Wave Alignment in Series-Text Attention

We provide an additional case study from the Sleep-EDF-20 dataset, focusing on time-series segments that exhibit prominent slow-wave activity. As illustrated in Figure 8, we visualize the co-attention scores between the EEG series and textual prompts at three different stages of model optimization.

In the early stage (a), the model assigns low and scattered attention across both modalities, indicating weak alignment. During intermediate training (b), attention becomes more concentrated, albeit not yet task-relevant. By the final stage (c), the network distinctly emphasizes time points associated

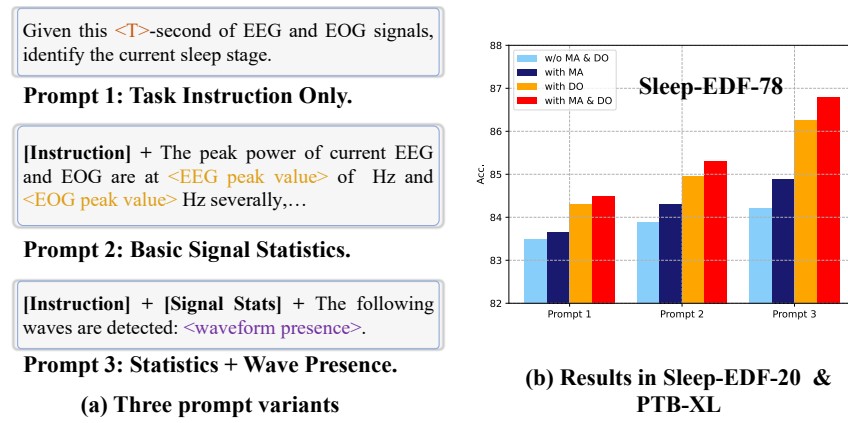

(a) Three prompt variants

(b) Results in Sleep-EDF-20 & PTB-XL

Figure 7: Prompt Robustness Analysis on Sleep-EDF-78.

with slow-wave peaks, and correspondingly shifts textual focus toward phrases referencing waveform patterns rather than general signal descriptions.

This progressive refinement highlights the model's capacity to learn localized waveform semantics from indicator-guided prompts. It also illustrates that our series-text alignment mechanism, enhanced by prompt design and optimization strategies, effectively bridges raw signal patterns and high-level textual representations—even under nuanced physiological phenomena like slow waves.

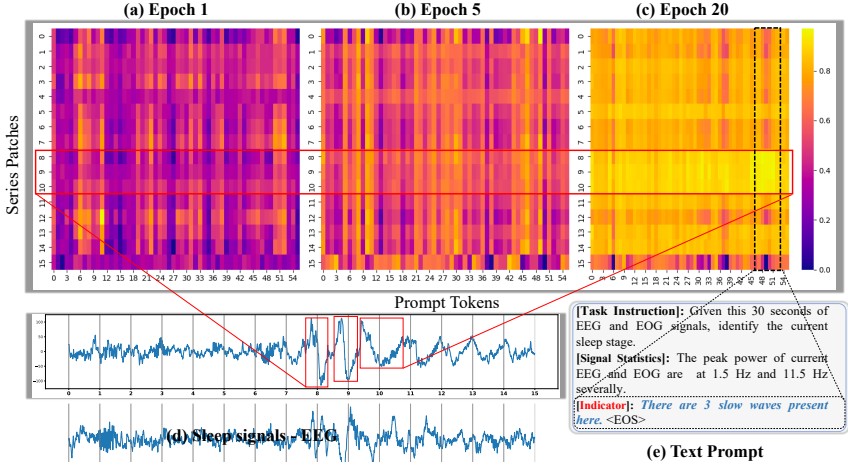

Figure 8: Visualization of co-attention dynamics for slow-wave segments. (a)-(c) depict attention maps at different training stages; (d)-(e) show corresponding series and prompt inputs.

# H  Limitations and Future Work

This work strives to reduce reliance on expert knowledge by employing semi-automated tools to extract signal indicators and selecting only a small subset of salient features rather than aiming for exhaustive domain coverage. While this approach improves scalability and practicality, it still implicitly assumes the availability of some domain knowledge—namely, a minimal understanding of what constitutes relevant cues or events within a given task. In domains where such priors are poorly defined or altogether absent, constructing meaningful prompts may remain challenging.

Looking ahead, a key direction is to enhance the generality of our framework by moving from instance-level cues to event-level schema abstraction. Rather than relying on manually defined signal indicators, future prompting could employ loosely defined task schemas that describe events or state changes, allowing LLMs to infer and internalize domain patterns in a self-supervised manner. This paradigm would reduce human intervention, enable rapid adaptation across physiological domains, and advance LLM-based temporal reasoning with minimal priors.

