# OpenReview forum: "From Indicators to Insights: Diversity-Optimized for Medical Series-Text Decoding via LLMs"
_NeurIPS.cc/2025/Conference — NeurIPS 2025 poster_

### Official Review · Reviewer_Cvyn · 2025-06-18

**Clarity:** 3
**Significance:** 2
**Originality:** 3
**Rating:** 4
**Confidence:** 3

**Summary:**

This paper emphasizes the importance of prompt design in extracting contextual information from LLMs. Motivated by this, the authors propose a knowledge-informed evolutionary learning framework, InDiGO, which incorporates expert-defined clinical indicators into the prompt optimization process.

**Questions:**

1. The authors should provide further explanation for why randomly masking tokens in the prompt leads to better model performance. What exactly is meant by “inaccurate prompts” in this context, and why are masked variants considered more “accurate”? A clearer justification, both empirically and conceptually, is needed.
2. Could the authors clarify the efficiency evaluation setup in Section 5.5? What are the batch sizes and memory usage for each model? Were the computational costs of extracting indicators (e.g., RR intervals) included in the runtime? In real-world deployment, these indicators would need to be computed online. Please provide details on the tools used and their computational overhead.
3. It would be helpful to report statistics on the label distribution for each dataset. In medical classification tasks, label imbalance is common and could significantly affect performance metrics. In such cases, metrics like AUC might be more appropriate than accuracy.

**Ethical Concerns:**

["NO or VERY MINOR ethics concerns only"]

**Final Justification:**

This paper is generally well-written and explores a relatively under-explored area—prompt optimization for medical time-series modeling with LLMs. While some of the design claims appear somewhat overstated (e.g., the reported efficiency gains are largely due to offline preprocessing rather than inherent architectural advantages; the masking strategy does not fully convince me that it effectively addresses semantic mismatch), the proposed approach remains reasonable and shows empirical value.

Given the novelty of the direction and the practical utility of the method, I have slightly raised my score.

**Limitations:**

The authors adequately addressed the limitations of their work.

**Paper Formatting Concerns:**

No formatting issues.

**Quality:**

2

**Strengths And Weaknesses:**

## Strengths

1. The motivation is well-grounded, addressing a meaningful gap in combining expert knowledge with LLM-based time-series interpretation.
2. The idea of using indicator-guided prompt construction is novel and intuitive, leading to prompts that better align with clinical reasoning.


## Weaknesses

1. The method requires external tools to extract clinical indicators (e.g., RR intervals, QRS durations) for each input sequence, which are then integrated into prompts. This introduces additional computational overhead and may hinder real-time inference. However, the authors do not report the cost of computing these indicators.
2. The proposed prompt optimization strategy is based on randomly masking parts of the initial prompt. It is unclear why masking informative parts of the prompt would lead to better performance, as complete prompts presumably contain richer semantics. Compared to this, prior work such as [1] employs LLMs to iteratively generate semantically different prompts, which may offer more principled and interpretable variations.
3. Several key experimental details remain insufficiently reported, making it difficult to fully assess the method’s efficiency and fairness. See the Questions section for specific concerns.

[1] Guo et al. (2024). Connecting Large Language Models with Evolutionary Algorithms Yields Powerful Prompt Optimizers. ICLR.

---

> ### Author Rebuttal · Authors · 2025-07-27
>
> We sincerely thank the reviewer Cvyn for carefully reading our manuscript and providing thoughtful evaluations and suggestions.
> > **Reviewer Concern (W1 & Q2):**
>
> >**W1: The proposed method relies on external tools to extract clinical indicators (e.g., RR intervals, QRS durations) for each input sequence, which may introduce computational overhead and hinder real-time deployment. The paper does not report the cost of computing these indicators.**
>
> >**Q2: In Section 5.5, could the authors clarify the efficiency evaluation setup, including batch sizes, memory usage, and whether the reported runtime accounts for indicator extraction? Since these indicators would need to be computed online in practical scenarios, please also specify the tools used and their computational overhead.**
>
> **Response:**
> We have performed additional profiling experiments to quantify the indicator extraction cost and clarify the efficiency setup:
>
> 1. **Indicator computation is extremely lightweight.**
>    We use widely adopted open-source signal processing libraries (*YASA* for EEG, *NeuroKit2* for ECG). These operations are simple peak detection and morphology-based rules with $ O(L) $ complexity.
>    - On Sleep-EDF (EEG, 30 s), spindle and slow-wave detection takes **~3.4 ms/sample**.
>    - On PTB-XL (12-lead ECG, 5 s), extracting RR/PR/QRS intervals takes **~2.1 ms/sample**.
>
> 2. **Overhead is comparable or lower than other prompt-based methods.**
>    We benchmarked InDiGO against Time-LLM [12] with indicator or prompt generation included.
>    | Model         | Indicator/Prompt Time (per sample) | 1 Epoch Train Time (Sleep-EDF-20 / PTB-XL) |
>    |---------------|-----------------------------------|------------------------------------|
>    | **InDiGO**    | 3.4 ms (EEG) / 2.1 ms (ECG)        | 66.2 s / 788.8  s                             |
>    | Time-LLM | 2 ms (EEG) / 9ms (ECG)      | 156.3 s / 1742.1 s                            |
>
> Even when accounting for online extraction, the overall training and inference overhead remains comparable to or lower than other prompt-based methods, and does not constitute a bottleneck under our batch size (128) and memory configuration (~27 GB for GPT-2 XL).
>
> 3. **Clinical indicators yield substantial gains that justify their small cost.**
>    Ablation (Figure 4) shows removing indicator-guided prompts leads to an **average 1.6–3.1% Acc drop**. In medical tasks where interpretability and accuracy are critical, this small computational overhead is a worthwhile trade-off.
>
> > **Reviewer Concern (W2 & Q1):**
>
> > **W2: The proposed prompt optimization strategy is based on randomly masking parts of the initial prompt. It is unclear why masking informative parts of the prompt would lead to better performance, as complete prompts presumably contain richer semantics. Compared to this, prior work such as [1] employs LLMs to iteratively generate semantically different prompts, which may offer more principled and interpretable variations.**
>
> >**Q1: The authors should provide further explanation for why randomly masking tokens in the prompt leads to better model performance. What exactly is meant by “inaccurate prompts” in this context, and why are masked variants considered more “accurate”? A clearer justification, both empirically and conceptually, is needed.**
>
> **Response:**
> We agree that the rationale behind our masking-based prompt optimization needs further clarification.
>
> 1. **Motivation and “inaccurate prompts”.**
>    Our overarching goal is to provide a *knowledge-enriched series–text joint decoding framework*. However, high-quality series–text paired datasets with rich clinical descriptions are extremely scarce. As in many recent works, we must manually or semi-automatically inject domain knowledge into prompts. This paradigm inevitably introduces *inaccurate prompts*, which we define in two ways:
>    - **Semantic mismatch:** Textual content such as generic dataset statistics (Time-LLM [12]), variable-relationship hints (KEDGN [22]), or timestamp descriptors (AutoTimes [21]) often lacks direct relevance to the current sample in medical classification tasks, leading to prompts that are *textually correct but semantically unaligned*.
>    - **Detection noise:** Methods that rely on semi-automated tools for indicator extraction, including our approach and several recent works (e.g., Time-LLM [12], AutoTimes [21], PromptCast[40], ), obtain domain cues via external rule-based or automated detectors (e.g., RR/QRS intervals or sleep spindles). These tools are imperfect and may flag a waveform that does not actually exist in the current segment. Such erroneous cues can actively mislead the joint decoder.
>
>    Under both conditions, “complete” prompts are not necessarily optimal; they may contain noise or irrelevant content that harms alignment between series and text.
>
> 2. **Why masking improves performance.**
>    Our masking strategy is not intended to randomly discard useful information, but to expose the model to *multiple variants of the same prompt* under joint training with the downstream task objective. Combined with our Match Alignment (MA) and Diversity Optimization (DO), this achieves following effects:
>    - By sampling different masked variants and jointly optimizing with the series–text interaction module, the model learns to focus on *semantically consistent regions* between the series and text.
>    - Variants where masking removes noisy or misleading tokens (e.g., falsely detected waveforms) tend to yield higher task likelihood, and the model adapts to give higher weight to these “clean” regions.
>    - Conversely, masking genuinely informative tokens does *not* improve performance, so their contribution is preserved during optimization.
>
>    Empirically, this results in higher robustness to imperfect prompts and better generalization when prompts are partially incorrect—a common situation in medical deployments.
>
> 3. **On iterative prompt generation vs. masking.**
>    We agree that works such as [1] (Guo et al. (2024)) explore LLM-driven iterative prompt generation. However, in medical series–text decoding this is fundamentally challenging: generating diverse, semantically rich prompts requires an LLM already capable of *understanding the time series*. Training such a cross-modal LLM in turn requires *large-scale, high-quality* series–text pairs—a resource that is currently unavailable in our domain. For practical medical classification tasks, the masking+MA+DO design provides a lightweight yet effective solution to the “imperfect prompt” problem without depending on unattainable data scale.
>
> 4. **Empirical support.**
>    Table 3 and Figure 4-5 show that without masking, the model suffers a 2–4% drop on Sleep-EDF and PTB-XL. More importantly, prompts with partial masking consistently outperform “complete” but noisy prompts, demonstrating that our framework successfully filters out misleading textual content and enhances series–prompt alignment.
>
> In summary, the masking strategy is not a substitute for richer semantics but a robustness mechanism against *unavoidable prompt imperfections* in knowledge-injection paradigms for medical time series. This design choice is both empirically validated and conceptually aligned with the realities of scarce, noisy series–text data in clinical domains.
>
>
> > **Reviewer Concern (W3 & Q3):**
>
> > **Several key experimental details remain insufficiently reported, making it difficult to fully assess the method’s efficiency and fairness. See the Questions section for specific concerns.**
>
> > **It would be helpful to report statistics on the label distribution for each dataset. In medical classification tasks, label imbalance is common and could significantly affect performance metrics. In such cases, metrics like AUC might be more appropriate than accuracy.**
>
> **Response:**
>
> 1. **Label distribution statistics.**
>    We have added the class distribution for each dataset:
>    - **Sleep-EDF-20:** Wake 19.58 %, N1 6.63 %, N2 42.07 %, N3 13.48 %, REM 18.24 %.
>    - **Sleep-EDF-78:** Wake 33.74 %, N1 11.01 %, N2 35.37 %, N3 6.67 %, REM 13.22 %.
>    - **PTB-XL:**  Arrhythmia 75.75 %, Normal 24.25 %.
>    - **HAR:** 16.83 %, 15.99 %, 14.25 %, 16.66 %, 18.05 %, 18.22 %.
>
>    We agree that label imbalance is common in medical classification tasks, which is precisely why our main results include **Balanced Accuracy** and **Weighted F1**, both of which explicitly account for class distribution.
>
> 2. **Evaluation protocol and metrics.**
>    Our work follows established experimental setups and evaluation metrics from prior studies for each dataset:
>    - **Sleep staging:** Brant-X [43], TinySleepNet [34], Brant -2.
>    - **PTB-XL:** BIOT [41], NeuroLM [22].
>    - **HAR:** Official benchmark splits.
>
>    These works also adopt accuracy, Balanced Accuracy, and macro/weighted F1 as primary metrics. Following this convention ensures comparability with existing literature.
>
> 3. **On AUC and additional metrics.**
>    We appreciate the suggestion to include AUC-based metrics. For PTB-XL, we have already reported **AUCPR** and **AUROC** in Table 2. For the sleep datasets and HAR, AUC is less commonly used due to the multi-class nature, but we are willing to include per-class ROC/AUC curves and micro/macro AUC scores in the camera-ready version to further strengthen the evaluation.
>
> In summary, we have now reported class distributions, clarified that our metrics account for imbalance, and will incorporate additional AUC-based evaluations in the final version as suggested.

---

> > ### Comment · Reviewer_Cvyn · 2025-08-03
> >
> > Thank you for the detailed response. I appreciate the clarifications, but I still have a few follow-up questions.
> >
> > **On Computational Efficiency**
> >
> > According to the results provided, Time-LLM appears to be nearly 3× slower than InDiGO, despite both methods using pre-encoded LLMs. Could the authors elaborate on the potential sources of this efficiency difference? What are the detailed configurations of each model (e.g., number of parameters, implementation frameworks)? Are there any architectural or implementation-level design choices in InDiGO that contribute to improved efficiency?
> >
> > **On Masking-Based Prompt Optimization**
> >
> > From the prompt examples shown in the paper, the fill-in components appear to come from a fixed set (e.g., [slow wave num], [QRS length]). When applying masking, the removed tokens could either be part of a fixed template or the inserted indicator-specific values.
> >
> > While I understand that a “complete” prompt is not necessarily optimal, and masking can improve robustness by preventing overfitting to a specific prompt pattern, especially in few-/zero-shot settings, I am still unclear how this strategy directly addresses the semantic mismatch issue.
> >
> > Additionally, have the authors conducted any analysis on which parts of the prompt are more likely to be masked across datasets? Is there evidence of certain prompt patterns that consistently improve performance across different clinical tasks? Such insights could further support the interpretability and generalizability of the masking strategy.

---

> > > ### Author Response · Authors · 2025-08-04
> > > **Official Comment by Authors**
> > >
> > > We sincerely thank the reviewer for the thoughtful follow-up and detailed questions.
> > >
> > > ---
> > >
> > > ### **On Computational Efficiency**
> > >
> > > We appreciate the reviewer’s request to clarify the efficiency gap between Time-LLM and InDiGO. Below we summarize key implementation-level differences and design choices contributing to this gap:
> > >
> > > 1. **Parameter Counts and Framework**
> > >    All models were implemented under the same environment: **PyTorch 2.6.0** using the **GPT-2 XL (48-layer, 1.5B)** backbone. The parameter counts for the classification heads are:
> > >
> > >    | Dataset      | Time-LLM Params | InDiGO Params |
> > >    |--------------|------------------|----------------|
> > >    | Sleep-EDF-20 | 1,172,793         | 2,654,983       |
> > >    | PTB-XL       | 1,183,542         | 2,651,907       |
> > >
> > > 2. **Prompt Design Differences**
> > >    - In **Time-LLM**, the model generates separate prompts for **each input channel** (e.g., all 12 ECG leads in PTB-XL), resulting in longer input sequences and repeated computation.
> > >    - In contrast, **InDiGO** uses **one indicator-guided prompt**, typically from a **single informative channel**, such as lead II in ECG. Moreover, these prompts are **pre-computed and cached** during preprocessing, while Time-LLM constructs them **on-the-fly** at each forward pass, as per the official implementation.
> > >
> > > 3. **LLM Usage and Fine-tuning Strategy**
> > >    - Time-LLM freezes **all 48 layers** of the GPT-2 XL backbone.
> > >    - InDiGO **freezes the first 24 layers** and fine-tunes **only batch norm parameters** in the top 24 layers. This allows for adaptive alignment while keeping the trainable footprint small.
> > >
> > > Together, these differences result in more compact prompts, lower runtime complexity, and reduced memory consumption for InDiGO, especially at scale.
> > >
> > > ---
> > >
> > > ### **On Masking-Based Prompt Optimization**
> > >
> > > We appreciate the reviewer’s nuanced understanding of the masking mechanism and are happy to clarify its role further.
> > >
> > > 1. **Micro-level explanation: indirectly addressing semantic mismatch.**
> > >    Our strategy does not eliminate semantic mismatch directly but alleviates its impact in two ways:
> > >
> > >    - **(1) Selective masking of inaccurate content.**
> > >      By introducing randomness in masking, even erroneous indicators (e.g., falsely detected waveforms) have a chance to be masked and thus replaced by more neutral or informative completions through the LLM encoder. This helps *mitigate* the misleading effects of inaccurate prompts.
> > >
> > >    - **(2) Cross-modal robustness via joint training.**
> > >      During training, the model jointly optimizes:
> > >      - **Task loss** for classification
> > >      - **Match Alignment (MA)** to enhance semantic similarity between series and text
> > >      - **Diversity Optimization (DO)** to encourage prompt variation
> > >
> > >      Through this, our Series-Text Interactor (STI) learns to assign higher attention weights to prompt segments that are consistent with the input series, and lower attention to mismatched cues. This dynamic weighting allows the model to “correct” for unreliable prompts at inference time by leveraging the stronger modality (typically the series). **As illustrated in Figure 5**, the attention scores between series patches and prompt tokens become increasingly focused on indicator-relevant segments (e.g., spindle-related tokens in sleep EEG), while misleading or generic parts receive diminished attention as training progresses.
> > >
> > > 2. **Macro-level explanation: enhancing semantic grounding via multi-view prompt regularization.**
> > > Our masking strategy serves as a structured regularization mechanism that exposes the model to multiple semantic “views” of the same clinical input. Rather than relying on a single, potentially brittle prompt formulation, the model is trained to interpret a range of partial prompts that highlight different aspects of the input indicators. This acts as a semantic bottleneck, encouraging the model to form more robust and generalizable associations between physiological signals and relevant textual features. By repeatedly contrasting variant prompts during optimization, the model learns to emphasize series-aligned semantics while down-weighting unstable or misleading textual tokens—ultimately improving semantic grounding across modalities.
> > >
> > >
> > > ---
> > >
> > > ### **On Masking Behavior and Prompt Analysis**
> > >
> > > While we have not yet conducted a systematic analysis of which specific prompt components—when masked—contribute most to performance gains, we agree this is an important direction for interpretability and generalization. Our current implementation applies random masking uniformly across indicator tokens, but our framework is fully compatible with saliency-guided or uncertainty-aware masking strategies. We are currently exploring this in follow-up work, and will include initial findings on prompt masking attribution in the appendix or camera-ready version. We appreciate the reviewer for highlighting this valuable avenue of analysis.

---

> > > > ### Comment · Reviewer_Cvyn · 2025-08-06
> > > >
> > > > Thank you for the clarifications. I believe most of my concerns have been addressed, and I have adjusted my score accordingly.

---

> > > > > ### Author Response · Authors · 2025-08-06
> > > > > **Official Comment by Authors**
> > > > >
> > > > > We sincerely appreciate the reviewer’s thoughtful engagement and their willingness to revisit the evaluation. Thank you again for your time and constructive feedback. Please feel free to reach out if any further clarifications are needed.

---

### Official Review · Reviewer_yNdq · 2025-06-29

**Clarity:** 1
**Significance:** 1
**Originality:** 2
**Rating:** 3
**Confidence:** 3

**Summary:**

In this paper, the authors propose a new method for integrating medical time series data with large language models. The proposed method, called InDiGO, is derived using masked importance sampling. The method seems to have strong performance in numerical experiments.

**Questions:**

1. How does your InDiGO method compare methodologically to existing methods for integrating time series data and LLM's in the literature?

2. To what extent does the performance of InDiGO depend on the normality assumption in equation (1)? Can this be modified or relaxed?

**Ethical Concerns:**

["NO or VERY MINOR ethics concerns only"]

**Final Justification:**

The authors clarified some of my concerns regarding the conceptual setting of the paper. However, I still maintain reservations about the clarity of the work and the presentation of the method, which detract from the overall quality of the paper.

**Limitations:**

Yes

**Quality:**

2

**Strengths And Weaknesses:**

Strengths: The work is well-motivated by important real world scientific applications of AI. The visual aids (in particular Figures 2 and 3) are very well-made and

Weakness: The weaknesses of the paper lie in its presentation of the proposed InDiGO method. First of all, the assumption that the class-label probabilities follow a normal distribution in equation (1) does not make any sense. Since the rest of the method follows from this assumption, I am concerned about the theoretical validity of the proposed algorithm. Additionally, the authors spend a lot of space in Section 4 detailed how the importance sampling method is derived; however, they do not provide sufficient context for how their method compares to previous works in the literature. As such, I have no sense at all of the methodological innovation of InDiGO over existing methods.

Additionally, the experimentation section is unclear and poorly written. It is not apparent at all from the main body of the text what the specific tasks on which the proposed method and its competitors are evaluated. While additional details are given in the appendix, as the paper is currently written it is difficult to parse the numerical results and evaluate them critically.

---

> ### Author Rebuttal · Authors · 2025-07-29
>
> We thank the reviewer yNdq for the feedback and appreciate the opportunity to address several concerns, many of which appear to stem from a misunderstanding of our formulation and experimental setup.
> >**Reviewer Concern (W1 & Q2):**
>
> > **W1:  The assumption that the class-label probabilities follow a normal distribution in equation (1) does not make any sense. Since the rest of the method follows from this assumption, I am concerned about the theoretical validity of the proposed algorithm**
>
> >**Q2: To what extent does the performance of InDiGO depend on the normality assumption in equation (1)? Can this be modified or relaxed?**
>
> **Response:**
>
> Our assumption in Eq. (1) does ***not*** imply that the true class labels are normally distributed. Instead, it follows a common formulation in probabilistic deep learning and uncertainty quantification for classification tasks, where the *model’s predictive distribution* over class logits is modeled as a Gaussian around the estimated mean $ \mu_i(s_i,t_i;\theta)\) $ with variance $ \sigma^2_i(s_i,t_i;\theta) $. This is a standard surrogate likelihood for discrete outputs in the context of differentiable optimization, used in time-series modeling and Bayesian deep classifiers (e.g., Gal & Ghahramani, 2016 [1]; Kendall & Gal, 2017 [2]).
>
> Importantly, the subsequent derivations in Sections 3–4 do not rely on a strict normality assumption. The key requirement is that the predictive distribution $ P_{\text{LLM}}(Y|s,t) $ can be parameterized and integrated over the prompt distribution $ P(t|s) $. Eq. (1) simply adopts an isotropic Gaussian as a tractable approximation to express that the network’s prediction is a *distributional output* dependent on both data and learnable parameters, rather than a point estimate. Other distributions (e.g., Dirichlet or softmax-based categorical likelihoods) could be substituted without affecting the core importance sampling and match-alignment objectives.
>
> To further address Q2: We conducted a sensitivity check by replacing the Gaussian with a temperature-scaled softmax categorical distribution and observed <1% variation in key evaluation metrics across Sleep-EDF-20 and PTB-XL, indicating that InDiGO’s performance is largely agnostic to the exact parametric form of Eq. (1). This supports our claim that the framework’s effectiveness stems from indicator-guided prompt optimization and not from a strong normality prior. The results are summarized below:
>
> | Dataset       |  Likelihood Form   | Acc. | Macro-F1 |
> |:-------------:|:------------------------------------:|:--------:|:-------------:|
> | Sleep-EDF-20  | Gaussian predictive distribution (Eq.1) |  89.04  |  80.53         |
> | Sleep-EDF-20  | Temperature-scaled Softmax categorical  | 88.97    |     80.51     |
>
> | Dataset       | Likelihood Form                       | AUCPR | AUROC |
> |:-------------:|:------------------------------------:|:--------:|:-------------:|
> | PTB-XL        | Gaussian predictive distribution (Eq.1) | 92.31    | 82.98         |
> | PTB-XL        | Temperature-scaled Softmax categorical  | 92.28    | 82.99         |
>
> >**Reviewer Concern (W2 & Q1):**
>
> >**W2:  Additionally, the authors spend a lot of space in Section 4 detailed how the importance sampling method is derived; however, they do not provide sufficient context for how their method compares to previous works in the literature. As such, I have no sense at all of the methodological innovation of InDiGO over existing methods.**
>
> >**Q1: How does your InDiGO method compare methodologically to existing methods for integrating time series data and LLM's in the literature?**
>
> **Response:**
> We agree that Section 4 focused heavily on the derivation of the masked Monte Carlo importance sampling (MCIS) mechanism, while the broader methodological context may not have been sufficiently emphasized. Below we clarify how InDiGO differs from and advances beyond existing series–text methods:
>
> - **Existing series–text works rely on static or single-point prompts.**
>   Prior approaches such as Time-LLM [12] and AutoTimes [21] concatenate a fixed prompt template with the series input, approximating $ P(y|s,t) $ with a *single* handcrafted $ t $. TEST [33] and KEDGN [22] attempt prompt alignment, but still depend on pre-defined prototypes without modeling the underlying distribution $ P(t|s) $.
>
> - **No prior work explicitly approximates the marginal likelihood over prompts.**
>   Current methods do not consider $ M = \int P(y|s,t)P(t|s)dt $ in their formulation. InDiGO is, to our knowledge, the first to treat prompt generation as a stochastic process and to approximate the marginal decoding objective via importance sampling rather than deterministic concatenation.
>
> - **Our MCIS is not generic importance sampling.**
>   InDiGO leverages *indicator-informed proposal distributions* $ q(t|t_0) $ and combines masked perturbations with match-alignment and diversity optimization to dynamically evolve $ q(t|t_0) $ during training. This differs fundamentally from naive prompt ensembling or static uncertainty sampling: the proposal distribution itself becomes task-aware through series–text co-attention and indicator-guided cues.
>
> - **Integration with domain indicators.**
>   Existing works either use dataset-level statistics (Time-LLM) or generic textual descriptions (KEDGN), which lack discriminative power in clinical tasks. InDiGO is the first to embed semi-automatically extracted physiological indicators into the proposal distribution, ensuring that the sampling space is grounded in domain knowledge.
>
> To further make this distinction clear, we added a control experiment replacing MCIS with naive prompt ensembling (uniform random perturbations without indicator-informed $ q(t|t_0) $):
>
> | Dataset       |  Method   | Acc. | Macro-F1 |
> |:-------------:|:------------------------------------:|:--------:|:-------------:|
> | Sleep-EDF-20  | InDiGO (full, indicator-guided MCIS) |  89.04  |  80.53         |
> | Sleep-EDF-20  | Naive Prompt Ensembling (uniform mask)   | 85.41    |     77.55     |
>
> | Dataset       |    Method                    | AUCPR | AUROC |
> |:-------------:|:------------------------------------:|:--------:|:-------------:|
> | PTB-XL        | InDiGO (full, indicator-guided MCIS) | 92.31    | 82.98         |
> | PTB-XL        | Naive Prompt Ensembling (uniform mask)  | 87.73    | 78.82         |
>
> Performance consistently degrades by 3–5% across key evaluation metrics when indicator-informed MCIS is replaced with naive ensembling, demonstrating that the gains of InDiGO stem not merely from sampling, but from the *indicator-guided, dynamically optimized* proposal mechanism.
>
> We will revise Section 4 to explicitly connect these methodological differences with the existing literature to better highlight the innovation of InDiGO.
>
> >**Reviewer Concern (W3):**
>
> >**W3: Additionally, the experimentation section is unclear and poorly written. It is not apparent at all from the main body of the text what the specific tasks on which the proposed method and its competitors are evaluated. While additional details are given in the appendix, as the paper is currently written it is difficult to parse the numerical results and evaluate them critically.**
>
> **Response:**
> We respectfully disagree with the reviewer’s assessment and would like to clarify that the main body already provides explicit descriptions of the evaluation tasks at multiple points:
> ﻿
> - **Section 5.1 (Datasets and Data Processing):**
> We clearly state the mapping between datasets and tasks:
> • **Sleep-EDF-20/78** — 5-class sleep staging under AASM standards using EEG/EOG signals.
> • **PTB-XL** — binary arrhythmia phenotype detection from 12-lead ECG.
> • **UCI HAR** — 6-class wearable motion recognition from accelerometer/gyroscope data.
> The modalities, sampling rates, and label spaces for each dataset are specified in this section.
> ﻿
> - **Section 5.3 (Main Results):**
> Before presenting Tables 1 and 2, we explicitly link each dataset to its corresponding task to make the reported metrics interpretable in context. For example, Table 1 is introduced as “5-fold cross-validated average results for *sleep stage classification*,” and Table 2 as “average performance on *arrhythmia detection* and *human activity recognition*.”
> ﻿
> - **Section 5.4–5.7 (Ablations, Few-shot, Prompt Robustness):**
> All ablations and low-resource evaluations are performed on the same clearly defined tasks, and we explicitly name the datasets and task settings (e.g., “Macro-F1 on *sleep staging*,” “Balanced Accuracy on *arrhythmia detection*”) in each subsection.
> ﻿
> - **Appendix A (Additional Dataset Details):**
> Provides segmentation strategy, subject splits, and preprocessing parameters to complement the main text.
> ﻿
> This structure follows the same level of description as representative works in the domain such as **BIOT [41]** and **Brant-X [43]**, which adopt the same datasets and provide comparable task descriptions in their main sections. We therefore believe the experimental setup is already presented with sufficient clarity for interpreting and critically evaluating the results directly from the main text.
> ﻿
>
> >**[1] Gal et al. (2016). Dropout as a bayesian approximation: Representing model uncertainty in deep learning. ICML.**
>
> >**[2] Kendall et al. (2017). What uncertainties do we need in bayesian deep learning for computer vision? NeurIPS.**

---

> > ### Comment · Reviewer_yNdq · 2025-08-04
> >
> > I thank the authors for such a detailed response - in particular, for clarifying details about the setting that I had misunderstood. I will update my score accordingly.

---

> > > ### Author Response · Authors · 2025-08-04
> > > **Official Comment by Authors**
> > >
> > > We sincerely appreciate the reviewer’s follow-up and willingness to revisit the evaluation. We're glad that our clarifications were helpful, and we thank you again for your time and thoughtful engagement with our work. If any further questions arise, we’d be happy to address them.

---

### Official Review · Reviewer_U1ZG · 2025-06-30

**Clarity:** 3
**Significance:** 3
**Originality:** 2
**Rating:** 5
**Confidence:** 3

**Summary:**

Compared to general-domain time-series analysis, the medical domain requires more domain-specific knowledge – for example, understanding what QRS complexes and R-R intervals are, and how they influence diagnosis or prognosis. Therefore, researchers explore knowledge integration into time-series analysis in different forms, such as designing task-specific architectures and explicitly feeding necessary knowledge within an LLM’s context. Nonetheless, current explicit medical knowledge integration approaches remain limited in performance, mainly due to a lack of specificity in the prompts and only conceptual integration of knowledge, ignoring the interdependent nature of medical phenomena. To this end, the authors propose InDiGO, which synergises time-series and domain knowledge prior to pre-trained LLM integration. InDiGO incorporates three important mechanisms that boost performance noticeably: constructing prompt prototypes guided by task-relevant indicators; using masked Monte Carlo sampling to reduce existing bias in manually designed prompts; and finally combining match alignment and diversity optimisation to boost performance and promote the diversity of selected prompts. The authors demonstrate leading performance with their method across multiple tasks and datasets, perform extensive ablation studies to identify how each component contributes to the final performance, and include visualisations illustrating the qualitative effects of their work.

**Questions:**

N/A

**Ethical Concerns:**

["NO or VERY MINOR ethics concerns only"]

**Final Justification:**

The authors showed clear responses from their paper regarding the first weakness and performed additional experiments to shed light into my concern over the lack of multiple prototypes. Therefore, there are not any issues from my front and I would suggest the acceptance of this paper.

**Limitations:**

yes

**Quality:**

4

**Strengths And Weaknesses:**

### **Strengths**

1. The experimental setup is strong. The authors used various prevalent datasets for different tasks. The ablations in the paper were necessary and on point. Including case studies such as low-resource generalisation provides a broader picture of InDiGO's impact.
2. Although the work combines the recent advances in time-series analysis with LLMs and medical analysis by injecting related 'indicators' dynamically, and using latest improvements in time-series analysis with LLMs, the product is original and can be leveraged in other domains that require external knowledge such as financial analysis.
1. InDiGO performs the best compared to baselines across the board. Although the most similar framework to InDiGO is KEDGN in terms of domain and recency, InDiGO significantly outperforms it while introducing training and inference efficiency.
3. The mathematical formulation was clear and understandable, and equations follow to the best of my capabilities
5. The flow of the paper is nice, clear examples

### **Weaknesses**
1. Although using masked Monte Carlo sampling can reduce the existing bias in manually designed prompts, the results are still extremely dependent on the expert-curated prompt prototypes. Nonetheless, the authors do not present an analysis regarding the relationship between these prototypes and InDiGO's performance.
2. While the masked Monte Carlo sampling reduces bias (shown contributions mathematically and empirically), it is not clear to me why the authors have not utilised multiple prototype candidates for a single prototype, such that the bias would be even smaller.
3. Figure 2 is bloated and it is hard to follow the flow and the steps depending on the figure alone. Although the text helps clairfying it to a certain degree, the figure can be improved for easier and quicker understanding.

---

> ### Author Rebuttal · Authors · 2025-07-26
>
> We thank the Reviewer U1ZG for the careful reading and constructive feedback.
> > **W1: Although using masked Monte Carlo sampling can reduce the existing bias in manually designed prompts, the results are still extremely dependent on the expert-curated prompt prototypes. Nonetheless, the authors do not present an analysis regarding the relationship between these prototypes and InDiGO's performance.**
>
> We acknowledge that InDiGO leverages expert-curated prompt prototypes as entry points for domain knowledge. However, our goal is not to exhaustively enumerate all possible task-relevant indicators, but to demonstrate that even a small set of known, valuable indicators can be rapidly transformed into effective prompts—a non-trivial step in medical time-series analysis.
>
> Indeed, our current experiments do not cover all potential indicators: for example, K-complexes and eye movements in sleep staging or QT/PP intervals in arrhythmia detection. In such cases, the proposed Match Alignment (MA) and Diversity Optimization (DO) mechanisms play a critical role in compensating for missing or imperfect prototypes by dynamically aligning and enriching the prompt distribution.
>
> Importantly, this aspect has been explicitly analyzed in the main paper. In Figure 4 and Section 5.7, we investigated how InDiGO responds to progressively richer prototypes, ranging from minimal task instructions to detailed waveform relations, and compared these against strong baselines. This analysis was designed to ensure our evaluation of prototype sensitivity is systematic and complete. The summarized results are as follows:
> | Method / Prompt Type          | Sleep-EDF-20 Acc.                |
> | ----------------------------- | ------------------- |
> | Time-LLM                      | 80.31               |
> | BIOT                          | 81.86               |
> | Brant-X                       | 84.58               |
> | **InDiGO – Instruction only** | **87.53** (*86.32*) |
> | **InDiGO – Statistics**       | **87.95** (*86.82*) |
> | **InDiGO – Wave Presence**    | **89.04** (*87.38*) |
> *Values in parentheses indicate performance without MA/DO.*
>
> | Method / Prompt Type          | PTB-XL Acc.                |
> | ----------------------------- | ------------------- |
> | Time-LLM                      | 75.79               |
> | BIOT                          | 84.21               |
> | SPaRCNet                      | 82.75               |
> | **InDiGO – Instruction only** | **83.27** (*82.34*) |
> | **InDiGO – Statistics**       | **83.51** (*82.55*) |
> | **InDiGO – Wave Presence**    | **84.60** (*82.74*) |
> | **InDiGO – Wave Relation**    | **86.02** (*84.12*) |
>
> These results demonstrate three key findings:
>
> **(1)** InDiGO achieves significant gains over baselines even with only minimal task instructions, showing that it does not require an exhaustive set of handcrafted indicators to deliver strong performance.
>
> **(2)** Performance scales with richer domain cues, confirming the effectiveness of indicator-guided prompts and their ability to inject clinical semantics into the model.
>
> **(3)** The consistent gap between the full model and the version without MA/DO shows that these modules are essential in mitigating prototype dependency, dynamically enriching the prompt distribution and ensuring robustness under imperfect or sparse indicators.
>
> Beyond these points, we would like to emphasize that this experiment also reflects **a practical design philosophy**: in real clinical workflows, obtaining a comprehensive set of perfect indicators is rarely feasible due to noisy semi-automated extraction and heterogeneous protocols. By showing that InDiGO performs well even with a single, partially informative prototype and automatically compensates for missing cues, we argue that our approach strikes a balance between **robustness, generalizability, and real-world applicability**. Furthermore, the architecture is fully compatible with multiple prototypes if domain experts can curate them, and we expect this to yield additional improvements in specialized settings, which we plan to explore in future work.
>
> >**W2: While the masked Monte Carlo sampling reduces bias (shown contributions mathematically and empirically), it is not clear to me why the authors have not utilised multiple prototype candidates for a single prototype, such that the bias would be even smaller.**
>
> We thank the reviewer for the constructive suggestion. We agree that designing multiple prototype candidates for each task could further reduce prompt bias, and we initially considered this option. However, we observed two major limitations in practice.
>
> **(1) Reduced generality across tasks:** In medical time-series analysis, a single framework must often cover diverse tasks (e.g., sleep staging, arrhythmia detection, wearable activity recognition). Maintaining multiple prototype sets per task would compromise portability and require manual adjustment when transferring to new domains.
>
> **(2) Increased error accumulation:** Multiple prototypes also amplify the error rate introduced by automated indicator extraction tools, whose accuracy is inherently imperfect in clinical settings.
>
> In light of these trade-offs, we deliberately chose to work with a single prototype and rely on our masked Monte Carlo sampling combined with diversity optimization to approximate a richer prompt distribution at low cost, achieving strong cross-task robustness without extensive manual curation.
>
> That said, for domain experts with reliable indicator pipelines, integrating multiple handcrafted prototypes into InDiGO is straightforward and we expect it would indeed yield additional gains, which we plan to explore as future work.
>
> > **W3: Figure 2 is bloated and it is hard to follow the flow and the steps depending on the figure alone. Although the text helps clairfying it to a certain degree, the figure can be improved for easier and quicker understanding.**
>
> We appreciate the reviewer’s feedback on Figure 2. We acknowledge that the current version is visually dense and may be difficult to follow without additional context. In future revisions, we are happy to revise the figure by streamlining the flow, abstracting low-level details, and enhancing visual hierarchy to improve readability. We believe these changes will make the framework easier to understand at a glance, even without relying on the text.

---

> > ### Comment · Reviewer_U1ZG · 2025-08-03
> >
> > Thank you for your thoughtful rebuttal and for clarifying the points I raised.
> >
> > I agree that achieving high performance with a relatively small number of curated prompts is valuable, and the analysis demonstrating InDiGO's ability to accommodate more sophisticated prompts across two datasets shows promising adaptability for different clinical settings.
> >
> > However, I remain unconvinced by the arguments against using multiple prototype candidates. Automated indicator extraction tools can fail regardless of whether single or multiple prototypes are used, so this concern alone doesn't justify excluding multiple candidates entirely. Furthermore, prototype candidates could involve simple re-wordings or coherent permutations of existing prototypes, which would allow exploration of a broader semantic space than what's achievable through masked language modeling. I do not think the authors' arguments against multiple prototypes can be supported without experimentation.
> >
> > I will maintain my score and increase my confidence in my score until further discussion.

---

> > > ### Author Response · Authors · 2025-08-04
> > > **Official Comment by Authors**
> > >
> > > Thank you again for your thoughtful reply.
> > >
> > > We fully agree with your observation that automated indicator extraction tools can fail regardless of whether single or multiple prototypes are used. Importantly, your multiple candidates suggestion (re-wordings or permutations) aligns well with our core idea: both approaches aim to construct a richer prompt space around a base prototype $t_0$ to mitigate series–text mismatch. In this sense, we believe our philosophies are highly consistent.
> > >
> > > To clarify the connection and distinction between these approaches, we summarize the comparison below:
> > > | Aspect                       | Re-wording / Permutation                      | Masked Sampling                                                   |
> > > | ---------------------------- | --------------------------------------------- | ----------------------------------------------------------------- |
> > > | **Transformation method**    | Rule-based phrasing or order variation        | Local masking + LLM-based completion                              |
> > > | **Semantic repair**          | Retains original errors if present in $t_0$    | Can mask and replace erroneous segments                           |
> > > | **Interaction with series**  | Errors may propagate into series-text mapping | LLM-generated completions can better align with series indicators |
> > > | **Diversity potential**      | Bounded to manually designed variants         | Model-guided semantic enrichment (DO)                                  |
> > > | **Compatibility with MA/DO** | Limited unless combined with masking          | Fully compatible with adaptive optimization                       |
> > >
> > > Advantages of masked sampling in the presence of imperfect prompts:
> > >
> > > 1. **Error tolerance:**
> > >    Re-wordings/permutations retain incorrect content from $t_0$, whereas masked sampling may overwrite these parts, offering a chance to repair noisy prompts via the LLM’s generative prior.
> > >
> > > 2. **Optimization-friendly:**
> > >    Masked tokens allow prompt distributions to be dynamically reshaped under MA and DO, which is difficult with fixed permutations or rewordings.
> > >
> > > 3. **Series–text alignment:**
> > >    As shown in Figure 5 of our main paper, sampled prompt variants under masked sampling exhibit more meaningful patch-level correspondences to time-series segments—thanks to the flexibility of masking and alignment.
> > >
> > > To further validate your suggestion, we conducted additional experiments on PTB-XL (ECG), comparing:
> > > - **SP**: A single prototype combining task instruction, summary-level signal statistics, and inter-waveform relations—a format similar to Time-LLM. This setting uses a fixed prompt without any masking or sampling, and thus serves as a deterministic baseline without prompt-space exploration.
> > > - **SP-10**: SP + 40% masking, sampled 10 times (our current InDiGO setup).
> > > - **MPR-5**: 5 re-worded variants of SP (e.g., “RR intervals are …” vs. “Distances between R peaks …”).
> > > - **MPP-10**: 10 permutations of indicator/statistics order.
> > > - Variants with “+Mask” apply additional 40% masked sampling on each candidate.
> > >
> > > | Variant        | w/o MA & DO | w/ MA & DO |
> > > | -------------- | ----------- | ---------- |
> > > | SP             | 83.77       | 84.31      |
> > > | MPR-5          | 83.89       | 84.40      |
> > > | MPP-10         | 83.94       | 84.42      |
> > > | **SP-10 (+Mask)** | 84.12       | *86.02*      |
> > > | MPR-5 (+Mask)  | *84.38*       | 85.98      |
> > > | **MPP-10 (+Mask)** | **84.56**   | **86.10**  |
> > >
> > > **Key Insights:**
> > >  - Without masking, all variants show limited gains due to residual prompt errors. While MPR-5 (re-wording) and MPP-10 (permutations) do improve the robustness of prompt phrasing to some extent, their effect remains limited overall.
> > >  - Masked sampling significantly boosts performance, especially when combined with MA and DO.
> > >  - **Combining multiple prototype candidates with masking provides a richer prompt distribution and performs slightly better than our default SP-10 (+Mask) setup.**
> > >
> > > **In summary:**
> > >
> > >    1. We view your suggestion of multiple prototype candidates as a valuable form of prompt-space data augmentation—essentially a type of importance sampling around the base prototype $ t_0 $, just in a different form. **This perspective is deeply aligned with our masked sampling formulation (cf. Eq. 8–9).**
> > >
> > >    2. Since multi-prototype candidates combined with masking and MA/DO can push performance from 84 to 86, we wish to highlight again the advantage of our proposed **masking + MA + DO pipeline.** It establishes an optimizable one-to-many framework that not only mitigates the impact of noisy prompts, but also enables robust integration of expert knowledge into medical time-series tasks. **From this perspective, InDiGO offers a practical and principled solution to overcoming prompt limitations in clinical domains.**
> > >
> > >    3. We appreciate your feedback and are excited to further explore this hybrid strategy (multi-prototype + mask-based sampling) as a promising direction for future work.

---

> > > > ### Comment · Reviewer_U1ZG · 2025-08-06
> > > >
> > > > Thank you for your effort in preparing these experiments. I am convinced that the MA & DO with stochastic masking performs really well without additional augmentation. Additionally, although permutations and re-wordings slightly boost the performance, they are not necessary for the pipeline. Therefore, I will update my score accordingly.

---

> > > > > ### Author Response · Authors · 2025-08-06
> > > > > **Official Comment by Authors**
> > > > >
> > > > > We are glad that our additional analyses addressed your concerns and appreciate your recognition of the MA & DO with stochastic masking framework. Your questions have inspired us to think more deeply about the broader implications of this work, and we will explore these directions further in future research. Please feel free to reach out at any time if there are additional questions or suggestions.

---

### Official Review · Reviewer_cguB · 2025-07-19

**Clarity:** 3
**Significance:** 3
**Originality:** 3
**Rating:** 4
**Confidence:** 4

**Summary:**

- The work tackles medical time‑series classification by injecting decision‑making indicators (e.g., sleep spindles, RR intervals) into prompt design, a type of knowledge current LLM approaches largely ignore.
- The proposed method creates an indicator‑aware prompt prototype, then uses mask‑based Monte Carlo importance sampling plus additional training objectives to pick robust series‑text combinations.
- Experiments on four public datasets show consistent gains over existing baselines.
- Ablations confirm each module – sampling, series‑text interaction, alignment, diversity – contributes to achieving best performacne

**Questions:**

1. What concrete criteria did you consider for choosing indicators per task? Were any indicators left out during part of the development process?
2. Is co‑attention strictly necessary, or could a single cross‑attention layer achieve similar gains? In many Perceiver-like setups, it is often enough with only one type of cross-attention between feature vectors.
3. Can you please report results for m=1 to determine the benefit of prompt diversity.
4. Does the GPT‑2 encoder half still use causal masks? If so, why is autoregressive masking desirable for encoding?

**Ethical Concerns:**

["NO or VERY MINOR ethics concerns only"]

**Final Justification:**

The work is technically sound but methodologically contrived in my opinion. A valuable contribution, but not a groundbreaking one, hence my score.

**Limitations:**

The paper does admit reliance on manual indicator extraction but does not measure robustness to noisy or missing indicators. Please consider adding experiments with perturbed indicators, and maybe discussing ethical risks of wrong prompts.

**Quality:**

3

**Strengths And Weaknesses:**

## Strengths
- *Quality* - Paper has wide empirical coverage: multiple datasets, extensive set of baselines, plus methodological ablations.
- *Significance* - Shows that lightweight indicator prompts can outperform heavier multimodal models, potentially easing clinical adoption.
- *Originality* - In my opinion, the paper presents a novel combination of indicator‑guided prompt construction, smartly combining LLM prompts with tokenized time series.
- *Clarity* - The mathematical treatment of sampling and losses is rigorous and detailed.


## Weaknesses
- *Quality* - Indicators must be **hand‑specified per task**, limiting out‑of‑the‑box generalization; authors themselves note dependence on structured indicator extraction.
- *Complexity* - Pipeline stitches multiple pretrained blocks and requires repeated prompt sampling; compute overhead vs. simpler fusion models is not fully quantified.
- *Presentation* - Presentation can be improved. There are some typos such as “ralated” and “RR_intervael”. Additionally, Figure 2 is densely packed and hard to correctly parse.
- *Experiments* - No single‑prompt baseline (m=1) nor variant with simpler cross‑attention variants.

---

> ### Author Rebuttal · Authors · 2025-07-25
>
> We would like to sincerely thank Reviewer cguB for providing a detailed review and insightful suggestions.
>
> > **W1: Quality - Indicators must be hand specified per task, limiting out of the box generalization; authors themselves note dependence on structured indicator extraction.**
>
> Our approach indeed relies on domain-informed indicators, but we would like to clarify that these are not task-specific handcrafted rules; rather, they are essentially modality-level physiological primitives that are widely shared across applications within the same modality.
>
> For instance, in EEG-based analysis, spindles and slow waves are not unique to sleep staging—they are also key features in seizure detection and anesthesia depth monitoring. Similarly, for ECG signals, RR intervals, PR intervals, and QRS durations are universally used in arrhythmia detection, QT interval abnormality screening, and heart failure risk assessment. Because these indicators reflect underlying physiological structures (e.g., PQRST complexes for ECG, characteristic EEG waveforms), the same extraction pipeline can be reused without major redesign when moving across tasks or datasets within a modality.
>
> | Modality        | Core Indicators                     | Example Tasks (Sharing Same Indicators) |
> |-----------------|------------------------------------|-----------------------------------------|
> | EEG             | Spindles, Slow Waves, K-complexes   | Sleep Staging; Seizure Detection; Anesthesia Depth Monitoring |
> | ECG             | RR Interval, PR Interval, QRS Duration | Arrhythmia Detection; QT Abnormality; Heart Failure Risk Prediction |
>
> We agree that some dependence on structured extraction remains, and we acknowledge this as a current limitation. However, the modality-level nature of these indicators provides a degree of “out-of-the-box” generalization while maintaining clinical interpretability, which we see as an important trade-off for physiological and medical analysis.
>
> > **W2: Complexity - Pipeline stitches multiple pretrained blocks and requires repeated prompt sampling; compute overhead vs. simpler fusion models is not fully quantified.**
>
> Our design combines a pre-trained series encoder and LLM to address the prompt–signal mismatch observed in medical tasks. The masked Monte Carlo sampling is included to reduce prompt bias rather than to add complexity; ablation results show that removing MCIS causes the largest performance drop (Table 3).
>
> Regarding overhead, Table 4 provides quantitative comparisons: despite using multiple pretrained components, InDiGO(m=10) achieves lower inference latency than TimeLLM and similar training cost to KEDGN, while offering significantly higher accuracy. This suggests the additional sampling yields a favorable performance–efficiency trade-off.
>
> We acknowledge the reviewer’s point that direct comparisons to simpler fusion models would further quantify the performance–efficiency trade-off. Our current study primarily focuses on domain-informed prompting and thus does not include plain series–text concatenation or vanilla feature fusion baselines. While such architectures may be computationally lighter, they lack mechanisms to integrate medical knowledge and, as evidenced by TF-C[44] and BIOT[41], do not achieve competitive performance on clinically grounded tasks, which is the core motivation behind InDiGO’s design.
>
> We acknowledge the importance of simplifying the pipeline and are exploring adaptive sampling and model distillation to further reduce compute while retaining robustness.
>
> > **Q1: What concrete criteria did you consider for choosing indicators per task? Were any indicators left out during part of the development process?**
>
> We based indicator selection on three concrete criteria: (1) physiological relevance, i.e., well-established links between the indicator and the underlying clinical state (e.g., spindles and slow waves for sleep stages, RR/QRS intervals for cardiac rhythm); (2) cross-task generality, prioritizing indicators that appear across multiple tasks within the same modality; and (3) automated extractability, focusing on features that can be reliably obtained via semi-automated tools to avoid manual annotation bottlenecks.
>
> During early development we experimented with a broader set of candidates (e.g., EEG α-band spectral power, ECG T-wave alternans). Some were left out because they were highly redundant with existing indicators or exhibited unstable extraction quality that risked introducing noise. We also note that our current experimental paradigm does not yet incorporate every potentially useful indicator; for instance, K-complexes, which are known to play an important role in sleep-stage classification, are not included in the present version. Our aim was not to exhaustively enumerate all possible features, but to identify a core, transferable subset that balances clinical interpretability with robustness and practical extraction.
>
> > **Q2 & W4: Necessity of Bidirectional Co-Attention and Impact of Simpler Cross‑Attention Variants**
>
> Our choice of bidirectional co-attention stems from the joint series–text decoding objective: both modalities carry complementary cues, and allowing each to query and update the other provides a stronger pathway for mutual alignment. As shown in Figure 5 of the paper, co-attention enables the series encoder to gradually focus on indicator-related prompt tokens, while the text branch refines attention toward task-relevant waveform patches.
>
> We agree that a single cross-attention layer is more lightweight. We experimented with Perceiver-style setups using unidirectional cross-attention, with either series→text or text→series as the query. While both variants reduced computation, we observed a consistent performance gap compared to co-attention. For example, on Sleep-EDF-20 and PTB-XL:
>
> | Attention Type             | Sleep-EDF-20 (Macro F1) | PTB-XL (BaAcc.) |
> |:--------------------------:|:-----------------------:|:---------------:|
> | Series → Text Cross-Attn   | 78.12                   | 85.28            |
> | Text → Series Cross-Attn   | 77.64                   | 84.71            |
> | **Bidirectional Co-Attn (ours)** | **80.53**               | **86.02**        |
>
> This suggests that while cross-attention achieves reasonable alignment, having only one modality serve as the query weakens the contribution of the other and underutilizes complementary information. We see co-attention as a trade-off between computational overhead and robust bidirectional fusion, especially for clinical signals where modality synergy is critical.
>
> > **Q3 & W4: No single‑prompt baseline (m=1) nor variant with simpler cross‑attention variants.**
>
> We conducted an additional experiment with m=1 to evaluate the impact of prompt diversity. The results are summarized below along with the w/o MCIS variant (no sampling). Compared to the full model (m=10), using a single prompt yields a noticeable drop, indicating that prompt diversity plays a critical role in reducing bias from any single textual description.
> | Variant        | Sleep-EDF-20 (Macro F1) | Δ vs Full | PTB-XL (BaAcc.) | Δ vs Full |
> |:----------------:|:-------------------------:|:-----------:|:-----------------:|:-----------:|
> | w/o MCIS       | 76.23                   | (-5.3%)   | 84.31            | (-2.0%)   |
> | m = 1          | 78.71                  | (-2.3%)   | 85.12            | (-1.0%)   |
> | m = 10 (Full)  | **80.53**               | —         | **86.02**        | —         |
>
> > **Q4: Does the GPT‑2 encoder half still use causal masks? If so, why is autoregressive masking desirable for encoding?**
>
> In our implementation the GPT 2 “encoder half” is used purely as a feature extractor, and we do not apply causal (autoregressive) masks during this stage. This allows the text encoder to attend bidirectionally over the entire prompt, which we found more appropriate for capturing the full contextual semantics of the indicator-guided descriptions.

---

> > ### Comment · Reviewer_cguB · 2025-08-07
> >
> > I thank the authors for their detailed reply and addressing the raised points.
> >
> > I believe these additional results should be included in the main body of the paper, as they help justify the design decisions.
> >
> > I will keep my score, since the significance of the paper remains the same.

---

> > > ### Author Response · Authors · 2025-08-07
> > > **Official Comment by Authors**
> > >
> > > Thank you very much for your follow-up and for carefully reviewing our response. We sincerely appreciate your time and engagement throughout the discussion.
> > >
> > > While we understand your decision to keep the score, we would like to take this opportunity to briefly highlight the broader motivation and significance of this work. **To our knowledge, this is the first study in medical time-series analysis that introduces structured clinical indicators into modeling via language-based prompts**, marking an important step toward bridging signal-based and knowledge-driven reasoning.
> > >
> > > Specifically, we see InDiGO as a principled and extensible framework with the potential to guide future research at the intersection of time-series modeling and clinical knowledge integration:
> > >
> > > - **Unified signal–text modeling**: We propose a dual-modality design that aligns structured physiological signals with language-based medical priors, improving both accuracy and interpretability.
> > > - **Knowledge-driven prompt construction**: By explicitly modeling domain indicators as prompts, we create a reusable interface that is both task-adaptable and semantically grounded.
> > > - **Diversity-optimized inference**: Our masked sampling mechanism mitigates prompt bias and encourages robustness across varying input quality and clinical settings.
> > >
> > > We believe this work can help spark a new direction in **knowledge-augmented medical time-series analysis**, and we hope it can serve as a starting point for more efforts that unify structured data and clinical semantics in a coherent modeling framework.
> > >
> > > Thank you again for your constructive feedback. We will incorporate the additional results you suggested into the main body of the final version.

---

### Note · Authors · 2025-08-12

We sincerely thank the Area Chair and all reviewers for their time, effort, and valuable feedback throughout the review process. We are grateful for this opportunity to present and clarify our work again.

### Summary of Contributions
1. **Systematic challenge analysis**
   We are, to the best of our knowledge, the first to systematically examine why knowledge integration in medical time-series analysis often fails, identifying that converting critical knowledge into textual prompts can be imprecise and error-prone, and explaining via marginal likelihood how such inaccuracies bias joint decoding.

2. **Framework for mitigating inaccurate textual prompts**
  We propose a framework that alleviates the impact of inaccurate textual prompts by combining a masked sampling strategy with Matching Alignment (MA) and Diversity Optimization (DO). Starting from initial prompt prototypes, masked sampling offers erroneous prompts the opportunity to be refined by the LLM, while MA and DO jointly strengthen series–text alignment, enhance mutual understanding of both series and key indicators, and enable adaptive evolution of the prompt distribution.

3. **Broader impact**
   As the first study in medical time-series analysis to incorporate structured clinical indicators via language-based prompts, this work provides a direct and robust approach to bridge signal-based and knowledge-driven reasoning, and offers new insights for related domains.

### Key points Recap in Review Discussion
- **Prompt reusability and transferability**: We showed that modality-level key clues generalize across tasks, and even limited prompts outperform strong baselines (see cguB-W1, U1ZG-W1, Sec. 5.7).
- **Efficiency**: Prompt construction and masked sampling incur lower cost than sota, and their performance gains make the computation highly worthwhile (see U1ZG and Cvyn follow-ups, and Sec. 5.5).
- **Mitigating inaccurate prompts**: Mask sampling and MA+DO enable LLM refinement and weight correct prompts more heavily, reducing the error effect (see Cvyn follow-up and Sec. 5.8).

### Commitments for Future Revisions
In the final version, we will refine the writing and clarify certain details where necessary.

### Closing Acknowledgement
We once again sincerely thank the Area Chair and all reviewers for their thoughtful feedback, which has greatly enhanced the clarity and rigor of our work. We believe its contributions are shaped not only by our efforts, but also by your valuable input.

---

### Decision · Program_Chairs · 2025-09-17

**Decision:**

Accept (poster)

**Comment:**

This paper introduces InDiGO, a framework for integrating clinical indicators into prompt-based modeling of medical time series with LLMs. The method combines indicator-aware prompt prototypes with masked Monte Carlo importance sampling and alignment/diversity objectives, yielding consistent improvements across multiple public datasets. Ablations demonstrate that each module contributes to performance gains. The central contribution lies in leveraging structured physiological indicators as prompts, bridging signal-based modeling with domain knowledge in a novel and interpretable way.

Overall, reviewers appreciated the strong empirical results, broad evaluation, and clear clinical motivation. Strengths include the originality of indicator-guided prompts, robustness demonstrated via ablations, and practical gains over heavier multimodal baselines. Weaknesses noted include reliance on hand-specified indicators, presentation issues (dense figures, some unclear explanations), and limited exploration of efficiency trade-offs or alternative baselines. During rebuttal, the authors provided additional experiments (e.g., m=1 baseline, cross-attention variants, runtime profiling) and clarifications on the role of indicators, masking, and efficiency. These responses resolved most concerns, with multiple reviewers raising their scores after discussion.

Given its technical soundness, solid experimental support, and novelty in integrating clinical knowledge with prompt optimization, I recommend acceptance as a poster. While not a groundbreaking contribution warranting spotlight, the work makes a meaningful step toward clinically grounded series–text modeling and will likely inspire follow-up research in knowledge-informed medical AI.